# Redefining Neural Operators in $d+1$ Dimensions

## Abstract

Neural Operators have emerged as powerful tools for learning mappings between function spaces. Among them, the kernel integral operator has been widely validated on universally approximating various operators. Although many advancements following this definition have developed effective modules to better approximate the kernel function defined on the original domain (with $d$ dimensions, $d = 1, 2, 3...$), the unclarified evolving mechanism in the embedding spaces blocks researchers' view to design neural operators that can fully capture the target system evolution.

Drawing on the Schrödingerisation method in quantum simulations of partial differential equations (PDEs), we elucidate the linear evolution mechanism in neural operators. Based on that, we redefine neural operators on a new $d + 1$ dimensional domain. Within this framework, we implement a Schrödingerised Kernel Neural Operator (SKNO) aligning better with the $d + 1$ dimensional evolution. In experiments, the $d + 1$ dimensional evolving designs in our SKNO consistently outperform other baselines across more than ten increasingly challenging benchmarks, ranging from the simple 1D heat equation to the highly nonlinear 3D Rayleigh–Taylor instability. We also validate the resolution-invariance of SKNO on mixing-resolution training and zero-shot super-resolution tasks. In addition, we show the impact of different lifting and recovering operators on the prediction within the redefined NO framework, reflecting the alignment between our model and the underlying $d + 1$ dimensional evolution.

## 1 Introduction

Neural Operators (NOs) (Li et al. (2021), Kovachki et al. (2023), Wu et al. (2023), Wu et al. (2024b)) parameterize mappings between two function spaces. Specifically, given inputs, they output function values corresponding with adaptable query coordinates in the continuous domain (with $d$ dimensions, $d = 1, 2, 3...$) where functions are originally defined. For operators mapping input to certain output functions defined on a $d$ dimensional domain, there are corresponding $d$ dimensional PDEs describing the governing law between variables of the target system. Once the in/output functions are chosen, the solution operator depends on the governing PDE type. A main challenge for representing different operators is that the corresponding $d$ dimensional PDE types vary across different systems.

Traditional numerical methods (Abdulle (2009); Bangerth (2008)) tackle this by directly converting $d$ dimensional governing PDEs into matrices using techniques like finite element methods and finite difference methods. However, such discretized signal evolving schemes are computationally prohibitive due to their reliance on a finer grid size to improve the solution accuracy (Larson & Bengzon (2013)).

Unlike traditional methods that rely on target system PDEs, common neural networks (NNs) universally learn the mapping between discretized input/output signal pairs on a grid in the $d$ dimensional domain (Hornik et al. (1989); Lagaris et al. (1998)), offering efficiency and adaptability to various governing PDE types. However, their predicted solution is limited by the grid size of signals (Obiols-Sales et al. (2020); Khoo et al. (2021)), suffering from the same problems as numerical methods, i.e., the trade-off between computational costs (training speed, memory allocation, etc.) and accuracy.

Since the problem for common neural networks lies in the size of signal grids, a different philosophy comes: learning operators rather than just matrices, because PDEs govern the evolution of continuous functions rather than discretized signals. A natural extension from matrix to operator is to define a

kernel integration to operate on $d$ dimensional functions corresponding to original signals, which has been widely used in modeling neural operators (Li et al. (2021); Liu-Schiaffini et al. (2024); Wu et al. (2024a); Li et al. (2020b)).

Specifically, these neural operators empirically introduce an embedding function and evolve it by $d$ dimensional kernel integral operators. Although recent advancements have developed effective modules, such as graph aggregation (Li et al. (2020a;b; 2025)), spectral convolution (Li et al. (2021); Wen et al. (2022)), and attention mechanism (Wu et al. (2024a); Xiao et al. (2024)), to better approximate kernel integral operators defined on the $d$ dimensional domain, little discussion for capturing the evolution within the embeddings of input/output functions was given:

**Problem 1.** *While many works (Kissas et al. (2022); Li et al. (2023b); Rahman et al. (2023)) have evaluated that such operator learning in an embedding space is empirically useful for representing various $d$ dimensional operators, what is the underlying mechanism of introducing an embedding space in PDE / Operator contexts?*

**Problem 2.** *Similar to previous matrix representations, current NO designs are still using discretized matrices to model the evolution within the embeddings. Can we similarly break the spell of unavoidable longer embeddings but heavier computational costs for better approximating?*

For problem 1, we try to answer: compared to the matrices corresponding to $d$ dimensional PDEs in common NNs, is there any PDE corresponding to kernel integral operators for the embedding functions in NOs? Here we refer to recent discoveries ((Jin et al. (2024))) in quantum simulation of PDEs: different $d$ dimensional linear PDEs can be converted to their $d + 1$ dimensional versions, as a Hamiltonian system in the discrete setting, by using simple warped phase transformations. While common NNs evolve $d$ dimensional signals through matrices, NOs evolve transformed $d + 1$ dimensional signals in the embedding space. Both are governed by PDEs with different descriptions of system mechanics. Demonstrations refer to Section 2.1.

For problem 2, we have located the bottleneck for current NOs: In previous common NNs, linear $d$ dimensional PDEs are represented by matrices. For the same system described by $d + 1$ dimensional PDEs, neural operators (Li et al. (2021; 2024b); Gupta et al. (2021)) introduce the kernel integral operator to capture the $d$ dimensional evolution, but still utilize matrices for evolving auxiliary embeddings in an Euclidean space. Following this framework, recent advanced methods (Li et al. (2023a); Wu et al. (2024a); Li et al. (2025)), especially transformer-based architectures, unavoidably increase the embedding size to better capture the embedding evolution but pay for their computational bill. For example, a compromised way is to use the multi-head to construct a larger block diagonal matrix rather than the complete one which better captures global dependencies during the embedding evolution.

Both problems indicate the same destination: a similar "operator rather than matrix" can be introduced to break the representation bottleneck of current NOs, aligning better with the underlying $d + 1$ dimensional evolution. Based on this idea, we firstly demonstrate the $d + 1$ dimensional evolving pipe of neural operators, and then redefine neural operators in the $d + 1$ dimensional domain (i.e., original $d$ dimensions + one auxiliary dimension). Following the redefined framework, we design the linear block of our model by combining kernel modules in the original domain and along the auxiliary dimension, and then implement the Schrödingerised Kernel Neural Operator (SKNO) within the new framework. Our main contributions in this work are summarized as follows:

- We redefine the current NO framework where embedding functions are defined on a new $d + 1$ dimensional domain according to the Schrödingerisation method (Jin et al. (2024)) in quantum simulation of PDEs, providing a new perspective to understand the evolving mechanism in neural operators, see Sec.2.

- We design our Schrödingerised Kernel Neural Operator (SKNO) within the $d + 1$ dimensional NO framework with linear blocks aligning with the general linear $d + 1$ dimensional evolution, utilizing $d + 1$ dimensional kernel integral operators with the aid of residuals, see Sec.3.

- We demonstrate the state-of-the-art performance of our $(d + 1)$-dimensional Linear Block and full model across a diverse set of more than ten benchmarks with varying levels of approximation difficulty. In addition, we conduct model-specific studies, including mixed-training, super-resolution, and ablation experiments, and further explore the impact of different lifting and recovering operators inspired by the observer effect in quantum mechanics.

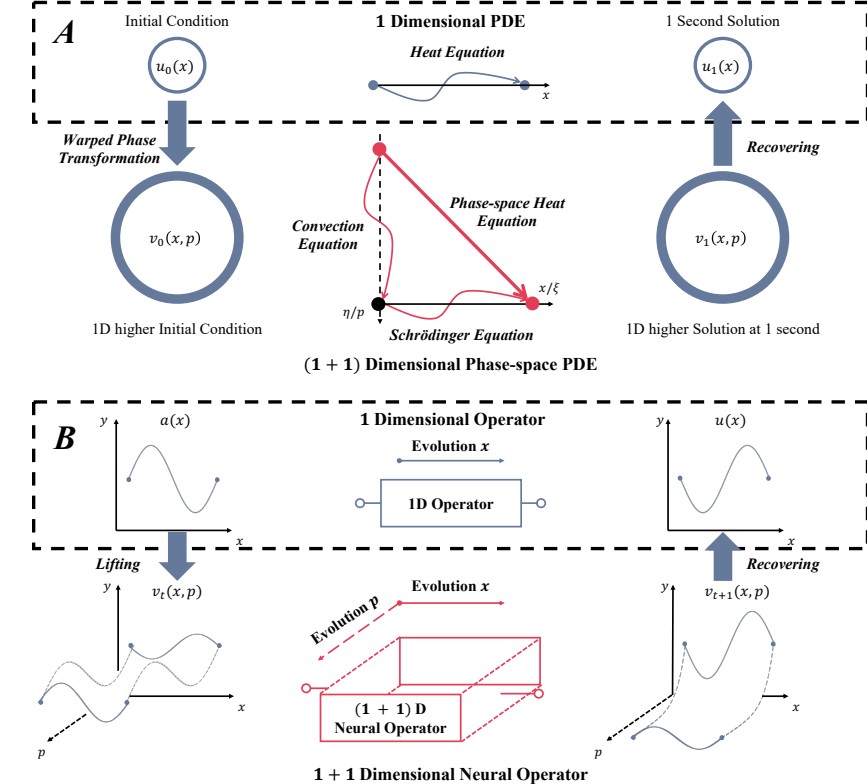

Figure 1: **A-Schrödingerisation**: Classical linear PDEs can be transformed into their $d + 1$ dimensional version (here $d = 1$ for visualization) by a warped phase transformation, e.g., heat equation $\rightarrow$ phase-space heat equation. And it can be represented by a group of Schrödinger or convection equations after taking the Fourier Transform on $p$ or $x$, see Remark 1. **B-Neural Operator**: Different $d$ dimensional linear operators can be similarly lifted to their $d + 1$ versions, and parameterized by the $d + 1$ dimensional Neural Operator which flexibly represents the evolution on both $x$ and $p$.

## 2 REDEFINING NEURAL OPERATOR IN $d + 1$ DIMENSIONS

### 2.1 DEMONSTRATIONS OF $d + 1$ DIMENSIONAL PIPES.

To simply clarify the $d + 1$ dimensional evolving mechanism related to problem 1, we demonstrate how to transform $d$ dimensional linear PDEs into their $d + 1$ dimensional versions with following examples in (Jin et al. (2022; 2023)):

**Example 1** (Heat Equation). *Consider the initial-value problem of the heat equation:*

$$\partial_t u = \partial_{xx} u, \qquad \text{(d dimensional heat equation)} \tag{1}$$

$$u(x, t = 0) = u_0, \qquad \text{(d dimensional input)} \tag{2}$$

*where $u = u(x, t)$, $x = (x_1, x_2, ..., x_d) \in \mathbb{R}^d$, $t \geq 0$. Let $d = 1$, which can be easily generalized to arbitrary $d$. Our goal is to construct a $d + 1$ dimensional pipe that evolves the initial condition $u_0 = u(x, t = 0)$ to the solution $u_1 = u(x, t = 1)$, shown in Fig.1 A.*

*Step 1: Warped Phase Transformation. Define a transformation from $u(x, t)$ to $v(x, p, t)$ by introducing an auxiliary dimension $p$:*

$$v(x, p, t = 0) = v_0(x, p) = e^{-|p|} u_0, \quad p \in (-\infty, \infty). \qquad \text{(d + 1 dimensional input)} \tag{3}$$

*Step 2: $d + 1$ Dimensional Phase-space Heat Equation. Combine Equ.1, 2 and 3, obtain:*

$$\partial_t v(x, p, t) = -\partial_p \partial_{xx} v(x, p, t). \qquad \text{(d + 1 dimensional heat equation)} \tag{4}$$

*Step 3: Recovering to $d$ Dimensional Solution. After evolving $v_0 \mapsto v_1 := v(x, p, t = 1)$ by Equ.4, Recover $u_1$ by integrating along $p$:*

$$u_1(x) = \int_{\infty}^{-\infty} \chi(p) v_1(x, p) dp, \qquad \text{(d dimensional output)} \tag{5}$$

*where $\chi(p)$ has multiple choices and depends on the introduced transformation form in Step 1. Here $\chi(p)$ can be formed as the step function or delta function.*

**Remark 1** (Fourier Transform of the Phase-space Heat Equation, Fig.1 A). *For Equ.3, if taking fourier transform on $p$ ($\mathcal{F}_p : p \to \eta$), one gets a group of uncoupled Schrödinger equations (operators), over all $\eta$; If taking fourier transform on $x$ ($\mathcal{F}_x : x \to \xi$), one gets a group of uncoupled convection equations (operators), over all $\xi$. Generally for $\boldsymbol{v}(t)$, discrete signals on $x$ and $p$ from $v(x, p, t)$, one gets a Hamiltonian system by taking the discrete fourier transform (DFT) on $p$. For Equ.3, the Hamiltonian system can be represented by further taking DFT on $x$:*

$$\frac{d}{dt}\hat{\tilde{\boldsymbol{v}}}(t) = i(D_\mu^2 \otimes D_\mu)\hat{\tilde{\boldsymbol{v}}}(t), \tag{6}$$

*where $\hat{\tilde{\boldsymbol{v}}}(t)$ is derived after taking DFT for $\boldsymbol{v}(t)$ on $p$ and $x$, and the same notation $D_\mu = diag(\mu_{-N_p/2}, ..., \mu_{N_p/2})$, which denotes a diagonal matrix with $N_p$ grid points in one dimension, is employed since little confusion will arise. Here, the diagonal coefficient complex matrix $i(D_\mu^2 \otimes D_\mu)$ is from taking DFTs on the $x$ and $p$ domain for $-\partial_p \partial_{xx}$ in Equ.3.*

**Example 2** (Advection Equation and Other Linear PDEs). *Similar process and result on advection equation as Example 1 and Remark 1 with introducing corresponding warped phase transformation $v_0 = sin(p)u_0$, $p \in [-\pi, \pi]$. Limited by pages, for more discussion of other PDE types, refer to (Jin et al. (2023)).*

According to (Jin et al. (2023); Jin & Liu (2024)), we could generally derive solutions for systems, which are governed by linear PDEs, through the $d + 1$ dimensional evolving process as Example 1. In this work, we follow this process to redefine our neural operator framework in $d + 1$ dimensions below, shown as Fig.1 *B*.

## 2.2 OPERATOR LEARNING IN $d + 1$ DIMENSIONS

**Problem Setup** Denote that $D_x \subset \mathbb{R}^d$ is a bounded, open set with $d$ dimensions. Given functions $a(x) \in \mathcal{A}(D_x; \mathbb{R}^{d_a})$ and $u(x) \in \mathcal{U}(D_x; \mathbb{R}^{d_u})$ with $x = (x_1, x_2, ..., x_d) \in D_x$, our goal is to approximate mapping operators $\mathcal{G}: a \mapsto u := \mathcal{G}[a]$. Here $\mathcal{A}$ and $\mathcal{U}$ are (suitable subsets of) Banach spaces, where functions $a$ and $u$ take values in $\mathbb{R}^{d_a}$ and $\mathbb{R}^{d_u}$ ($d_a, d_u \in \mathbb{N}$), respectively.

Similar to Example 1, we split our framework into three main components based on (Li et al. (2021)) for redefining the $d + 1$ dimensional neural operator framework:

**Lifting** We prepare the $d + 1$ dimensional function $v(x, p) \in \mathcal{V}(D; \mathbb{R})$ for system evolving in $d + 1$ dimensions, where $D = D_x \times D_p$. Given that we only have the input signal on the original domain, we design the lifting operator $\mathcal{P}$ with a separated form as Equ.3, which involves multiplying a learnable function $w(p)$, $p \in D_p$, $w \in \mathbb{R}^{d_a}$ whose values can be optimized by the back-propagation algorithm in neural implementations. The mathematical description of lifting operator $\mathcal{P}$ is:

$$v(x, p) = \mathcal{P}[a](x, p) = w^T(p)a(x). \tag{7}$$

$d + 1$ **Dimensional Evolution** To respond to the problem 2 in previous methods, we define the kernel integral operator on the $d + 1$ dimensional domain by introducing an auxiliary variable $p$. Define the $d + 1$ dimensional kernel integral operator with $\kappa : \mathbb{R}^{d+1} \to \mathbb{R}$:

$$\mathcal{K}[v](x, p) = \iint_{D_x \times D_p} \kappa(x, y, p, p') v(y, p') dy dp'. \tag{8}$$

Note that for non-linear cases, the evolution is usually modeled by iteratively stacking a linear block $\mathcal{L}$ and a non-linear block $\sigma$, which works well in $d$ dimensional NNs (Chen et al. (2018)) and $d + 1$ dimensional NOs (He et al. (2024)), shown in Fig.2. With the core component $\mathcal{K}$ in Equ.8, the implementation of $\mathcal{L}$ depends on specific model designs, see Sec.3.1.

**Recovering** After evolving $v(x, p)$, we take weighted integration on $D_p$ with a learnable function $\chi(p)$, $\chi \in \mathbb{R}^{d_u}$ to recover solution $u(x)$ back to the original domain. Define Recovering Operator $\mathcal{Q}$:

$$u(x) = \mathcal{Q}[v](x) = \int_{D_p} \chi(p) v(x, p) dp. \tag{9}$$

Combining the introduced blocks above, now we define our neural operator framework to approximate $\mathcal{G}$ in Sec.2.2 as follows:

**Definition 1** (Neural Operator in $d + 1$ dimensions). *Assume the $d$ dimensional solution operator $\mathcal{G}$ has at least one corresponding operator in $d + 1$ dimensions, define the framework below:*

$$\mathcal{G}[\boldsymbol{a}] = (\mathcal{Q} \circ \mathcal{M} \circ \mathcal{P})[a]$$
$$\approx (\mathcal{Q} \circ (\sigma_{L-1} \circ \mathcal{L}_{L-1}) \circ \cdots \circ (\sigma_0 \circ \mathcal{L}_0) \circ \mathcal{P})[a], \tag{10}$$

*where $v_{l+1} = (\sigma_l \circ \mathcal{L}_l)[v_l]$, $l = 0, 1, ..., L - 1$. Here, each $\mathcal{L}_l$ denotes a general learnable $d + 1$ dimensional linear operator, and $\sigma_l$ can be a non-linear activation function or a shallow MLP ($p$-dim-evolving operator + activation + $p$-dim-evolving operator).*

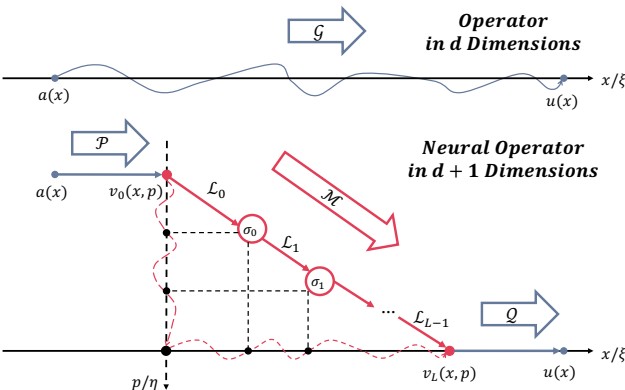

Figure 2: Neural Operator framework in $d + 1$ dimensions clarification. *Above*: Target $d$ dimensional operator $\mathcal{G}$, mapping input $a(x)$ to output $u(x)$; *Below*: Neural Operator $d+1$ dimensional framework, where the lifted $d + 1$ dimensional function $v(x, p)$ from input $a(x)$ evolves iteratively through linear and non-linear updates and finally recover to $d$ dimensional solution $u(x)$.

## 3 SCHRÖDINGERISED KERNEL NEURAL OPERATOR

In this section, we show implementation details of our proposed **Schrödingerised Kernel Neural Operator (SKNO)** within the redefined $d + 1$ dimensional framework, which is informed by the Schrödingerisation process in Sec.2. Sec.3.1 introduces the architecture and implementation of the SKNO, including the modules from preparation lifting to recovery measuring. Different from Sec.3.1 which mainly models the evolution in $D_x$, Sec.3.2 focuses on how SKNO better captures the evolution along the auxiliary dimension for its alignment with general linear PDEs.

### 3.1 MODEL ARCHITECTURE AND IMPLEMENTATION

An overview of the SKNO architecture is shown in Fig. 3. Except for the lifting and measuring modules, we implement the SKNO by constructing signal propagators with the kernel integration in Equ.8, and with the aid of residuals. Each of them propagates the $d + 1$ dimensional signal $\boldsymbol{v}_l$ with different inductive biases on global or local regions on $D_x$.

**Lifting and Recovering** For the lifting module in SKNO, we choose the simplest way to implement $w(p)$ in Equ.7 with a linear layer, which balances the representation flexibility with computational efficiency. From Example 1, *Step 3*, we have flexible implementation choices for $\chi(p)$ in Equ.9. For highlighting the power of our $d + 1$ dimensional evolving design, here we keep using MLP as the projection layer in other works, where the last linear layer implements the integration in Equ.9 as numerical quadrature. We also conducted comprehensive experiments and further explanations of different implementation choices for $\mathcal{P}$ and $\mathcal{Q}$ in Sec.4.2.

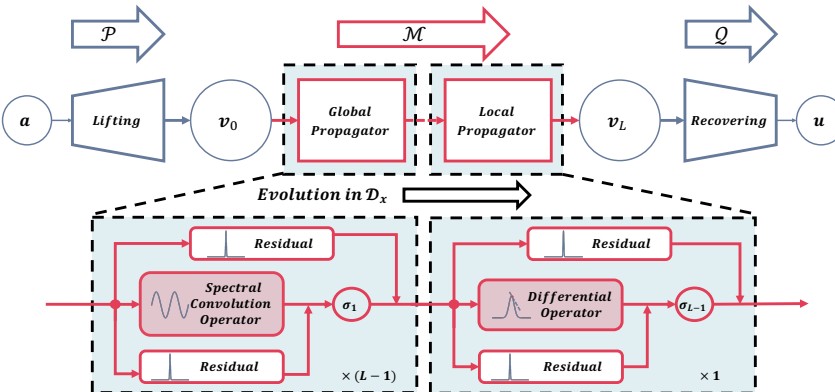

Figure 3: Overview of the Schrödingerised Kernel Neural Operator (SKNO) architecture. The input signal $a$ is lifted to a $d + 1$ dimensional signal $v_0$ through the preparation lifting module. Such a $d + 1$ dimensional signal evolves through $(L - 1)$ global propagators and one local propagator in $D_x$, capturing both global and local signal changes. The measuring module recovers the output signal $u$ from evolved $v_L$.

$d + 1$ **Dimensional Signal Propagation on $D_x$**   As illustrated in Fig.3, we employ $(L - 1)$ global propagators and one last local propagator for modeling the underlying $d + 1$ dimensional function evolution in $D_x$.

**Spectral Convolution Operator**   In the Schrödingerisation process (Jin et al. (2023)) like Example 1 and 2, different differential operators with respect to $x$ (e.g. $\partial_x, \partial_{xx}, ...$) can be represented by a complex diagonal matrix in Fourier domain of $x$ according to the properties of Fourier Transform.

Based on that, we leverage the implementation of spectral convolution operator in (Li et al. (2021)), which uses a learnable complex vector equivalent to a diagonal matrix in the Fourier domain of $D_x$, to parameterize the evolving operator related to $d + 1$ dimensional PDEs. A typical example is shown in Remark 1. In practice, the complex vector is truncated for working on different grid sizes:

$$\mathrm{SpectralConvOp}(\boldsymbol{v}_l) = F_x^{-1}(\mathrm{trun\_diag}(F_x(\boldsymbol{v}_l))), \tag{11}$$

where $F_x$ and $F_x^{-1}$ denote DFT and Inverse DFT (accelerated by Fast Fourier Transform (FFT) in practice). The $\mathrm{trun\_diag}()$ represents the element-wise multiplication between a learnable truncated complex vector and signal slices of $\boldsymbol{v}_l$ on the transformed grid $\xi$.

**Differential Operator**   Since the truncation in Sec.3.1 prefer to propagate signals in a relatively global scope, we compensate for possible local information lost by propagating the finite difference derivative of $x$ on the local region:

$$\mathcal{K}_{L-1}[v_{L-1}](x, p) = \frac{\iint_{D_{loc} \times D_p} \kappa_{L-1}(x, y, p, p')(v_{L-1}(y, p') - v_{L-1}(x, p'))dydp'}{\iint_{D_{loc} \times D_p} \kappa_{L-1}(x, y)(y - x, p')dydp'}, \tag{12}$$

where $D_{loc} \subset D_x$ denotes the local region around the query position $x$. In this work, the implementation of Equ.12 is based on the convolution layer in (Liu-Schiaffini et al. (2024)).

**Linear Block with Residuals**   With above two implementations of $d+1$ dimensional kernel integral operators in Equ.8, we specified the linear block $\mathcal{L}$ in Def.1 with two parts: kernel integral operator $\mathcal{K}_l$ and residual operator $\bar{\mathcal{W}}_l$:

$$\mathcal{L}_l[v_l](x, p) = (\mathcal{K}_l + \bar{\mathcal{W}}_l)[v_l](x, p), \tag{13}$$

where $\bar{\mathcal{W}}_l[v_l](x, p) = (\mathcal{I} + \mathcal{W}_l)[v_l](x, p) = v_l(x, p) + \int_{D_p} w_l(p, p')v_l(x, p')dp'$. Here, $\mathcal{W}_l$ is implemented by a shallow MLP and $\mathcal{I}$ is implemented by a residual connection.

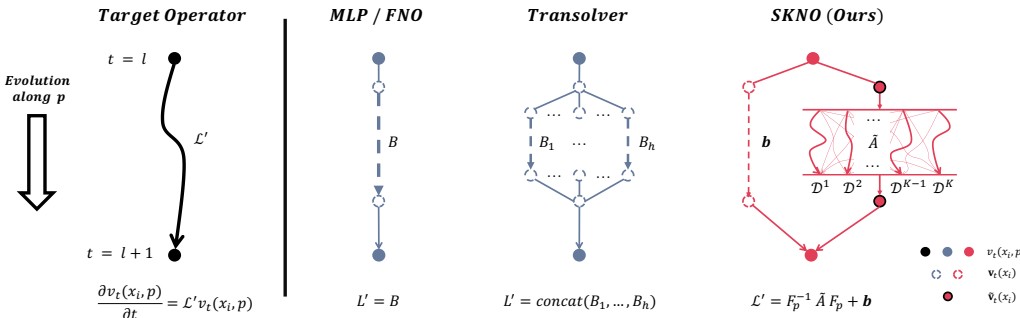

Figure 4: Evolution along the auxiliary dimension and neural operator implementations. *Left*: The target operator $\mathcal{L}'$ governs the evolution along $p$. *Right*: Compared with other methods using discretized matrices for each position of $p$, our model captures the target evolution with operator modules in both position and momentum spaces.

## 3.2 Evolution along the Auxiliary Dimension

In this section, we will show our model captures the evolution along the extra dimension with general linear PDEs:

$$\frac{\partial v_t(x_i, p)}{\partial t} = \left( \sum_{k=1}^{K} a_k(p) \frac{\partial^k}{\partial p^k} + b(p) \right) v_t(x_i, p) \tag{14}$$

where $x_i = (x_{i_1}, ..., x_{i_d})$, $i = (i_1, ..., i_d)$. Note that the same conclusion also works for $\xi_i$. And $K$ represents the highest order of the differential operator w.r.t $p$.

The evolution governed by Equ.14 is implemented by the forward finite difference method on $t = l$ (Let $\Delta t = 1$ to simplify the equation, which can be adjusted during learning.) as follows:

$$v_{l+1}(x_i, p) - v_l(x_i, p) = (\mathcal{K}_l + \mathcal{W}_l) [v_l](x_i, p), \tag{15}$$

which is exactly our linear block in Equ.13. For discrete signal along $p$, we implement the right hand side of Equ.15 by $F_p^{-1} \tilde{A} F_p + \boldsymbol{b}$, where $\tilde{A}$ denotes a learnable matrix in Fourier domain on $p$ (convolution theorem) and $\boldsymbol{b}$ corresponds to the discretized $b(p)$. As illustrated in Fig.4, while other methods should adjust their discretization size of $p$ to capture the evolving dynamics in Equ.14, our model integrates both weighted differential operator represented in Fourier domain on $p$ (SKNO, right) and bias term (SKNO, left) to align itself with general linear PDEs as Equ.14. By iteratively updating the $d + 1$ dimensional signal using the combination of these operator designs, our SKNO model aligns itself well with $d + 1$ dimensional PDEs to effectively offer improved scalability and adaptability over previous methods.

## 4 Experiments

### 4.1 General Setting

**Benchmarks** We evaluate our model on more than ten diverse benchmarks, covering a wide spectrum of PDE dynamics. To highlight the effectiveness of our evolving operator along the auxiliary dimension, we first test on simple linear dynamics, including the 1D Heat Equation (Shin et al. (2022)) and Advection Equation (Takamoto et al. (2022)). We then move to more challenging nonlinear and unstable regimes, conducting extensive experiments on the 1D Burgers Equation, 2D Darcy Flow (discontinuous coefficients), 2D incompressible Navier–Stokes (low viscosity $1e{-}5$) (Li et al. (2021); Wu et al. (2024a)), 2D Gray–Scott Reaction–Diffusion (Kassaï Koupaï et al. (2024)), and the highly unstable 3D Rayleigh–Taylor Instability (Ohana et al. (2024)). Additional benchmarks and detailed results are provided in Appendix C.

**Baselines and Configurations** We compare our model with various $d$-dimensional neural implementations, including DeepONet (Lu et al. (2021)), Fourier Neural Operator (FNO) (Li et al. (2021)), Convolutional Neural Operator (CNO) (Raonic et al. (2023a)), and transformer-based model

Table 1: Linear block $\mathcal{L}$ performance on learning operators governed by 1D Heat and Advection Equations.

Table 2: Ablation study for SKNO on 2D Darcy. The Linear Residual is $\bar{\mathcal{W}}_l$ in Linear Block $\mathcal{L}$, while the non-linear one is out of $\mathcal{L}$.

| Linear Blocks | Error |
|---|---|
| **1D Heat Equation** | |
| FNO (1 mode, 4 grid $p$) | 9.305e-1 |
| Transolver (1 slice, 4 grid $p$ / 2 heads) | 9.060e-1 |
| Transolver (4 slices, 4 grid $p$ / 2 heads) | 8.988e-1 |
| Transolver (4 slices, 4 grid $p$ / 1 head) | 8.987e-1 |
| **Ours (1 mode, 4 grid $p$)** | **2.637e-3** |
| **1D Advection Equation** | |
| Transolver (8 slices, 16 grid $p$ / 4 heads) | 7.118e-1 |
| Transolver (8 slices, 16 grid $p$ / 2 heads) | 6.984e-1 |
| Transolver (8 slices, 16 grid $p$ / 1 head) | 6.828e-1 |
| FNO (8 modes, 16 grid $p$) | 6.012e-2 |
| **Ours (8 modes, 16 grid $p$)** | **1.979e-2** |

| Configuration | Error |
|---|---|
| w.o. Double Res. | 1.363e-2 |
| w.o. Linear Res. | 8.284e-3 |
| w.o. Non-linear Res. | 7.621e-3 |
| w.o. $\tilde{A}$ along $p$ dim | 8.804e-3 |
| w.o. $\boldsymbol{b}$ along $p$ dim | 6.827e-3 |
| w.o. Global Propagators | 1.584e-1 |
| w.o. Local Propagator | 5.715e-3 |
| **Baseline (SKNO)** | **5.555e-3** |

Table 3: Main experimental results for the 1D Burgers, 2D Gray-Scott (GS), 2D incompressible NS and 3D Rayleigh-Taylor Instability. For 2D GS and NS, errors are measured within the next 10 time steps with autoregressive predictions.

| Benchmark | 1D Burgers Equation | | | | 2D Gray-Scott | | | |
|---|---|---|---|---|---|---|---|---|
| Model | DeepONet | FNO | Transolver | **SKNO** | DeepONet | FNO | Transolver | **SKNO** |
| Training Time | **2.07 min** | 6.86 min | 94.23 min | 15.51 min | **3.01 min** | 9.64 min | 234.96 min | 20.11 min |
| Rel. $L_2$ Error | 8.991e-2 | 6.479e-4 | 6.277e-3 | **5.475e-4** | 1.528e-1 | 2.425e-2 | 3.573e-2 | **1.298e-2** |
| | 2D Incompressible NS | | | | 3D Rayleigh-Taylor Instability | | | |
| | DeepONet | FNO | Transolver | **SKNO** | DeepONet | FNO | Transolver | **SKNO** |
| Training Time | 865.75 min | 773.33 min | 1263.33 min | **708.93 min** | **1.56 min** | 5.12 min | 135.03 min | 10.31 min |
| Rel. $L_2$ Error | 3.448e-1 | 1.280e-1 | 1.002e-1 | **8.717e-2** | 5.731e-2 | 5.219e-2 | 5.723e-2 | **4.471e-2** |

Transolver (Wu et al. (2024a)). All models are trained under their officially recommended optimal configurations, following the settings reported in their respective papers. For benchmarks such as 1D Burgers, 2D Darcy Flow, and 2D Gray–Scott, we train each model for 500 epochs with no physical prior regularization, using only relative $L_2$ loss. For the incompressible 2D Navier–Stokes, we ensure fairness by matching comparable training time scales, accounting for differences in training speed between transformer-based and other operator architectures. All experiments are run on a single *Nvidia GeForce RTX 4090* GPU. Complete configurations are provided in Tab.7.

## 4.2 MAIN RESULTS AND ANALYSIS

**Results** Our SKNO model consistently outperforms existing methods, achieving the lowest relative $L_2$ error with competitive training costs. Tab. 1 summarizes the performance of different linear blocks $\mathcal{L}$ across representative models: the Fourier Integral Operator in FNO, Physics-Attention in Transolver, and our redefined Spectral Convolution Operator. All modules are evaluated with their original lifting and recovering components, without additional positional encoding (see Appendix C). For the heat equation experiment, we predict the solution at $t = 1$ following Example 1, keeping the same setting as in practice with their mode or slice truncated. Note that these two parameterize the invertible aggregation on $D_x$. For the advection equation, we test these blocks in an autoregressive way to validate our proposed module's performance on concatenated input quantities and prediction tasks. Specifically, we calculate the prediction error here of the autoregressive prediction in the next 10 time steps from the last 10. While other modules are limited by their evolving design on $p$, ours breaks this bottleneck and stably well captures the $d + 1$ dimensional evolution. Across the benchmarks in Tab. 3, Tab. 11 and Tab. 13, SKNO achieves state-of-the-art (SOTA) performance, with training time comparable to FNO which is well-known for balancing speed and accuracy. Visualizations in Fig. 5 and Fig. 11 further demonstrate that SKNO excels in capturing sharp, local details, such as discontinuities at material interfaces and vorticity in low-viscosity fluids. For complete experimental settings, case studies, and complete results, see Appendix C.

Table 4: The error on 1D Heat and 2D Darcy with different $\mathcal{Q}$ implementations.

| $\mathcal{Q}$ | Delta | Step | Mean | Linear | MLP | MLP (Dropout) |
|---|---|---|---|---|---|---|
| | 1D Heat Equation | | | | | |
| Error | **2.538e-3** | **2.538e-3** | **2.538e-3** | **2.538e-3** | 2.637e-3 | 7.928e-3 |
| | 2D Darcy Flow | | | | | |
| Error | 5.752e-3 | 1.402e-2 | 5.717e-3 | 5.741e-3 | **5.555e-3** | 5.678e-3 |

Table 5: The error on 2D Darcy with different $\mathcal{P}$, $\mathcal{Q}$ combinations.

| $\mathcal{P}$ $\diagdown$ $\mathcal{Q}$ | Linear | MLP |
|---|---|---|
| Constant | 8.738e-3 | 8.791e-3 |
| Linear | 5.741e-3 | **5.555e-3** |
| MLP | 5.952e-3 | 6.614e-3 |
| MLP (Dropout) | **5.451e-3** | 5.845e-3 |

**Resolution Invariance**    A key advantage of neural operators is their resolution invariance. To verify this, we design novel experiments where training data of mixed resolutions are combined. Tab. 11 shows that SKNO consistently achieves the best accuracy, even under randomly mixed training resolutions, demonstrating robustness. In addition, we further evaluate super-resolution performance on the 1D Burgers equation, 2D Darcy flow, and the challenging ERA5 wind field prediction task. All models are trained on a fixed resolution and tested on lower and higher ones. As shown in Fig. 5, SKNO maintains a consistently low relative $L_2$ error across resolutions, significantly outperforming FNO on 1D Burgers. Other results in Tab. 10 and Tab. 12 indicate that competing models exhibit large error fluctuations when resolution increases, revealing their instability in generalizing across resolutions. In contrast, SKNO provides more stable and reliable predictions across varying scales.

**Different Lifting and Recovering Operators**    Motivated by the assertion that the preparation and measurement of quantum states influence the outcomes in quantum mechanics, we explore different lifting and recovering operators to test their impact on the output solution. Here, the lifted $d + 1$ dimensional function describes a quantum state as (Jin et al. (2023)). First, we fix $\mathcal{P}$ with a linear layer and vary the integration weights $\chi(p)$ for $\mathcal{Q}$. Tab. 4 shows that all recovering designs proposed in Example 1 perform well on the heat equation, where the Mean and Linear can also be properly scaled since the symmetry of $w(p)$. Amazingly, some predefined recovering operators can still work on complex Darcy cases, where the complete $p$ evolution provides detailed local changes on $D_x$ after the evolution on complete $p$, see Fig.10. Combining results from Tab. 4 and Tab. 5 on 2D Darcy, we find that for functions with complex topology on $D_x$, a continuous non-linear transformation on $d + 1$ dimensional signals, either after first lifting or before last recovering, can help it adjust the $d + 1$ dimensional propagating structure. The best performance with MLP-Linear combination is consistent with the MLP-Linear implementation in Transolver (Wu et al. (2024a)) for handling complex topology. What's more, comparing the Transolver performance in Tab.3 with ours in Tab.5, we also conclude that our $d + 1$ dimensional design captures the evolution better on 2D Darcy.

**Ablation**    We conduct a detailed ablation study on 2D Darcy (Tab. 2) to evaluate the contribution of different components. Removing both linear and non-linear residual layers causes a significant performance degradation, confirming their necessity for modeling complex $d + 1$ dimensional evolution. Similarly, omitting the $p$ differential operator in the Fourier domain $\tilde{A}$ or the bias term $b$ from the auxiliary dimension dynamics results in higher errors, where the introduced $\tilde{A}$ contributes more to the model prediction. Finally, we separate the contributions of the global and local propagators. The global propagator captures long-range dependencies, while the local propagator refines predictions in regions with sharp transitions, corresponding to the scale of rebounded errors after masking one of them and keeping another.

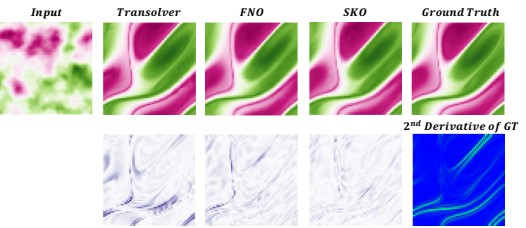 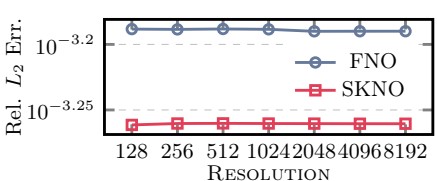

Figure 5: *Left*: Prediction, error, and second derivative of ground truth (GT) visualization for 2D Darcy; *Right*: Model performance on the Burgers, grid size ranging from 128 to 8192.

## 5 CONCLUSION, FUTURE WORK AND LIMITATIONS

In this work, we redefine a novel $d + 1$-dimensional framework for neural operators, inspired by the Schrödingerisation method that reformulates classical $d$-dimensional PDEs within a one-dimension-higher domain. Building on this foundation, we proposed the Schrödingerised Kernel Neural Operator (SKNO), which employs an efficient $d + 1$ dimensional kernel integral operator to flexibly capture the underlying evolution. Across more than ten benchmarks, SKNO achieves state-of-the-art accuracy with competitive computational efficiency. Furthermore, its demonstrated resolution invariance and comprehensive ablation studies confirm both the robustness and validity of the proposed design. Taken together, these results establish SKNO as a principled and effective extension of neural operator methods, while also motivating broader exploration of quantum-inspired operator learning frameworks.

As in previous neural operators (Li et al. (2021); Hao et al. (2023); Wu et al. (2024a)) and related theoretical works (Kovachki et al. (2021); Lanthaler et al. (2025)), a main open problem is how to theoretically quantify and compare the general approximation errors of different neural operators. In future work, we will seek to rebuild the theoretical foundations from the perspective of the redefined function space $\mathcal{V}(D; \mathbb{R})$ and explore whether the $(d + 1)$-dimensional formulation can help alleviate the current theoretical challenges.

## 6 ETHICS AND REPRODUCIBILITY STATEMENTS

This work does not involve human subjects, personal data, or other sensitive information. All experiments were conducted on publicly available or synthetically generated datasets. To ensure reproducibility, we provide detailed descriptions of the experimental setup, including model architectures, training procedures, and evaluation protocols. The code and datasets used in this study will be made publicly accessible upon publication. For readers, the code implementations and related configurations are included in the supplementary materials.

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

## A  TABLE OF NOTATIONS

A table of key notations is given in Table 6.

Table 6: Table of key notations.

| Notation | Meaning |
|---|---|
| **Problem Setup** | |
| $D_x \subset \mathbb{R}^d$ | Original domain where functions are defined with $d$ dimensions |
| $x \in D_x$ | Query position of in/output functions in the $d$ dimensional domain |
| $a(x) \in \mathcal{A}$ | Input function |
| $d_a$ | The length of concatenated input vectors |
| $u(x) \in \mathcal{U}$ | Output function |
| $d_u$ | The length of concatenated output vectors |
| $\mathcal{G}[\cdot]$ | Solution operator |
| $d+1$ **dimensional NO Framework** | |
| $p$ | The auxiliary dimension variable composing $d+1$ dimensional PDEs |
| $D_p$ | Domain where the auxiliary variable $p$ lives |
| $v(x, p)$ | Transformed function in the $d+1$ dimensional domain $D_x \times D_p$ |
| $\mathcal{P}[\cdot]$ | Operator lifts input signals into ones with grids in $D_x \times D_p$ |
| $\mathcal{L}[\cdot]/\mathcal{L}_l[\cdot]$ | The ($l$-th) learnable linear block capturing $d+1$ dimensional operators |
| $\sigma_l(\cdot)$ | The ($l$-th) element-wise activation function or 2-layer MLP |
| $\mathcal{Q}[\cdot]$ | Operator recovers evolved $d+1$ dimensional signals into outputs |
| **Others** | |
| $\hat{(\cdot)}$ | The original function/signal represented in Fourier domain of $x$ |
| $\tilde{(\cdot)}$ | The original function/signal represented in Fourier domain of $p$ |
| $\widehat{(\cdot)}$ | The original operator/matrix represented in Fourier domain of $x$ |
| $\widetilde{(\cdot)}$ | The original operator/matrix represented in Fourier domain of $p$ |
| $N_x$ | The number of grid indices on $D_x$ |
| $N_p$ | The number of grid indices on $D_p$ |
| $k_x$ | The number of grid indices on the transformed domain after aggregation on $D_x$ |
| $k_p$ | The number of grid indices on the transformed domain after aggregation on $D_x$ |

## B  STATEMENT ON LLM USAGE

In this paper, we used a large language model (LLM) to provide suggestions on word choice, grammar, and logical consistency. While some sentences were lightly rephrased by the LLM, all scientific content remains original and authored by us.

## C  EXPERIMENT SETTINGS AND RESULTS

**Loss**  Denote the parameterized neural operator as $\mathcal{G}_\theta$, we aim to approximate the solution operator $\mathcal{G}$ in Equation 10 by optimizing the model parameters $\theta \in \Theta$ through relative $L_2$ loss below:

$$\min_{\theta \in \Theta} \mathcal{L}(\theta) := \min_{\theta \in \Theta} \frac{1}{S} \sum_{s=1}^{S} \left[ \frac{\|\mathcal{G}_\theta[a_s] - u_s\|_2}{\|u_s\|_2} \right] \tag{16}$$

$$= \min_{\theta \in \Theta} \frac{1}{S} \sum_{s=1}^{S} \left[ \left( \frac{\sum_j (\mathcal{G}_\theta[a_s](x_j) - u_s(x_j))^2}{\sum_j (u_s(x_j))^2} \right)^{\frac{1}{2}} \right], \tag{17}$$

where $\Theta$, $S$, and $u_s = u_s(x)$ denote the parameter space, the number of function samples, and the $i$-th output function, respectively.

**Benchmarks and Results**  Benchmarks and the corresponding experimental results are summarized below:

1. *1D Heat Equation*

   Consider the 1D Heat Equation as follows:

   $$\partial_t u = c\partial_{xx} u, \tag{18}$$

   where $c = 0.05$ as in (Shin et al. (2022)). We generate 1200 samples (1000 for the training set, and 100 for testing) in a denser grid with size $2^{10} = 1024$, and down-sample them to size $2^8 = 256$. Our goal is to predict solution $u_1 = u(x, t = 1)$ from $u_0 = u(x, t = 0)$. Results are shown in Tab.1.

2. *1D Advection Equation*

   Consider 1D Advection Equation:

   $$\partial_t u = \beta \partial_x, \tag{19}$$

   where $\beta$ is set with 1. We use the data from (Takamoto et al. (2022)) with 2000 samples of $u_0 = u(x, t = 0)$. Our goal is to predict the next 10 time steps from the last 10 of the solution u(x, t). The number of training and testing samples is 1000 and 200, respectively. Results are shown in Tab.1

3. *1D Burgers Equation*

   The 1D Burger's Equation on the unit torus is formed as (Li et al. (2021)):

   $$\partial_t u + u\partial_x u = \nu \partial_{xx} u, \tag{20}$$

   where $\nu = 0.1$ with periodic boundary conditions. Our goal is to build a solution operator to predict the solution $u(x)$ at $t = 1$. The training and testing grid size is set with $2^7 = 128$. Results are listed in Tab.3.

4. *2D Gray-Scott System*

   Consider a pair of coupled reaction-diffusion PDEs as follows:

   $$\partial_t u = D_u \Delta u - uv^2 + F(1 - u),$$
   $$\partial_t v = D_v \Delta v - uv^2 - (F + k)v, \tag{21}$$

   where $D_u = 0.2097$, $D_v = 0.105$ denote the diffusion coefficients respectively for $u, v$. And $F = 0.03, k = 0.062$ are the reaction parameters. We follow the data generation in (Kassaï Koupaï et al. (2024)) to make a training set with 200 samples of $u, v$ evolution and a testing set with 20 in a $32 \times 32$ size grid with periodic boundary conditions. Our goal is to predict the next 10 time steps from the last 10 of $u, v$. See Tab.3 for results.

5. *2D Shallow-Water Equations*

   The 2D shallow-water equations govern the evolution of fluid flows under the assumption that horizontal length scales are much larger than the vertical depth. They can be expressed as the following system of hyperbolic PDEs:

$$\partial_t h + \nabla h\mathbf{u} = 0, \tag{22}$$

$$\partial_t h\mathbf{u} + \nabla \left( \mathbf{u}^2 h + \frac{1}{2}g_r h^2 \right) = -g_r h \nabla b, \tag{23}$$

where $h = h(x,t)$ denotes the fluid depth, $\mathbf{u} = \mathbf{u}(x,t)$ is the 2D velocity field, $g_r$ is the (reduced) gravitational acceleration, $b = b(x)$ represents the bathymetry characterizing the bottom topography. For this benchmark, we employ the dataset from (Takamoto et al. (2022)), which is generated under Neumann boundary conditions. To assess resolution-invariant stability during training, models are trained on 500 samples with resolutions randomly drawn from $\{32, 64, 128\}$ and evaluated on 300 test samples at the highest resolution 128. The prediction task is to forecast the water depth over the next 10 time steps, given the previous 10. See Tab.11 for results.

6. *2D Darcy Flow*

The 2D Darcy Flow is a second-order elliptic equation as follows:
$$-\nabla \cdot (a \nabla u) = f, \tag{24}$$
where $f = 1$. Our goal is to approximate the mapping operator from the coefficient distribution function $a(x)$ to the solution function $u(x)$ with zero Dirichlet boundary conditions (Li et al. (2021)). The results are based on models trained on 1000 training samples, which are downsampled to a size of $(85 \times 85)$, and tested on 100 samples. Results are listed in Tab.11.

7. *2D Incompressible Navier-Stokes Equations*

The 2D incompressible Navier-Stokes (NS) Equations in vorticity form on a unit torus are shown below:
$$\partial_t \omega + u \cdot \nabla \omega - \nu \Delta \omega = f, \tag{25}$$
$$\nabla \cdot u = 0. \tag{26}$$
Here $\nu = 1e{-}5$ and $f = 0.1(sin(2\pi(x_1 + x_2)) + cos(2\pi(x_1 + x_2)))$. Our goal is to predict the next 10 time steps from the last 10 ($d_a = d_u = 10$) of the solution $w(x,t)$ with periodic boundary conditions on the 2D grid with size $64 \times 64$ (Li et al. (2021)). All models in Table 3 are trained on 1000 training samples and tested on 200 test samples. Results are listed in Tab.3.

8. *3D Compressible Navier–Stokes Equations*

The 3D compressible NS equations describe fluid dynamics with density changes, given by:
$$\partial_t \rho + \nabla \cdot (\rho \mathbf{v}) = 0, \tag{27}$$
$$\rho(\partial_t \mathbf{v} + \mathbf{v} \cdot \nabla \mathbf{v}) = -\nabla p + \eta \triangle \mathbf{v} + (\zeta + \eta/3)\nabla(\nabla \cdot \mathbf{v}), \tag{28}$$
$$\partial_t \left[ \epsilon + \frac{\rho v^2}{2} \right] + \nabla \cdot \left[ \left( \epsilon + p + \frac{\rho v^2}{2} \right) \mathbf{v} - \mathbf{v} \cdot \sigma' \right] = 0, \tag{29}$$

where $\mathbf{v} = \mathbf{v}(x,t)$, $\rho = \rho(x,t)$, and $p = p(x,t)$ denote the velocity vector field, fluid density (mass per unit volume), and pressure, respectively. We consider the challenging transonic flow benchmark from (Takamoto et al. (2022)), with shear and bulk viscosity coefficients $\eta = \zeta = 1 \times 10^{-8}$, and internal energy density $\epsilon = \frac{3p}{2}$. The viscous stress tensor $\sigma' = \sigma'(\mathbf{v}; \eta, \zeta)$ is determined by the velocity field under the Newtonian fluid assumption. Our objective is to predict the velocity, density, and pressure fields three time steps ahead, given the previous three steps ($d_a = d_u = 3 \times 5 = 15$). The simulations are performed on a downsampled 3D grid of size $32 \times 32 \times 32$ with periodic boundary conditions. The initial conditions consist of turbulent velocity fields with uniform mass density and pressure. All benchmark models are trained on 500 samples and evaluated on 100 test samples. See Tab.11 for results.

9. *3D Rayleigh-Taylor Instability*

The Rayleigh–Taylor (RT) instability occurs when a heavier fluid overlies a lighter one under the influence of gravity, which is extremely sensitive to perturbations of the interface and governed by the following PDEs:

$$\partial_t \rho + \nabla \cdot (\rho \boldsymbol{u}) = 0, \tag{30}$$
$$\partial_t (\rho \boldsymbol{u}) + \nabla \cdot (\rho \boldsymbol{u} \boldsymbol{u}) = -\nabla p + \nabla \cdot \boldsymbol{\tau} + \rho \boldsymbol{g}, \tag{31}$$
$$\nabla \cdot \boldsymbol{u} = -\kappa \nabla \cdot \left( \frac{\nabla \rho}{\rho} \right). \tag{32}$$

where $\rho = \rho(x,t)$ and $\boldsymbol{u} = \boldsymbol{u}(x,t)$ denote the density and the velocity vector field, respectively. $p$ is the pressure, $\boldsymbol{\tau} = \boldsymbol{\tau}(\boldsymbol{u}, \rho; \nu)$ is the eviatoric stress tensor with kinematic viscosity $\nu$, $\boldsymbol{g}$ represents the gravity, and $\kappa$ is the coefficient of molecular diffusivity. A key dimensionless parameter in the RT simulations is the Atwood number $A = (\rho_h - \rho_l)/(\rho_h + \rho_l)$, which quantifies the density contrast between the two fluids. For example, miscible fluids with greater density contrasts exhibit stronger instabilities. In this work, we adopt the dataset with $A = 0.25$ provided in (Ohana et al. (2024)) to train a solution operator $vec(\rho, \boldsymbol{u}; t = t_0) \mapsto vec(\rho, \boldsymbol{u}; t = t_0 + 20)$ which evolves the system from rest states (with small perturbations) to mixed states after 20 time steps. The models are trained on 220 mapping pairs and tested on 40 pairs from distinct trajectories, using a downsampled grid of size $32 \times 32 \times 32$. Horizontal periodic boundary conditions and impermeable vertical walls are imposed. See Tab.3 for results.

10. *ERA5 Wind Field Prediction*

To compare the baseline models in a global context of spatio-temporal dynamics, we evaluate their performance on predicting the next-hour ERA5 global wind field (Hersbach et al. (2020)). All models are trained using data from the previous four days to predict the next-hour wind field at a grid size of $512 \times 512$, and are tested on data from a new day with the same grid size. See Tab.13 for results.

We emphasize the broader learning difficulty spectrum of the benchmarks we selected. As shown in Fig. 5 (Li et al. (2021)) which illustrates the spectral decay of truncation modes, our chosen benchmarks 6 and 7 contain significantly more high-frequency modes than simpler benchmarks such as 3 or other incompressible NS benchmarks with $\nu > 1 \times 10^{-4}$. To further evaluate the capability of these efficient and lightweight neural operators in capturing complex non-linear dynamics in 3D space, we include two additional, extremely challenging benchmarks: 8 and 9. Moreover, we also compare the performance of all baseline models in the rapid prediction of global wind fields, providing a comprehensive evaluation across both highly nonlinear and large-scale spatio-temporal dynamics.

**Baselines** The principle of our experiment configurations for different baselines is to keep the best configuration provided in their papers or codes, while ensuring fairness and not exploding the *Nvidia GeForce RTX 4090* GPU memory (limited to 24564MiB). Detailed configurations of all baselines in different experiments are shown in Table 7. And some explaination notes are listed below:

- Mainly two types of Positional Features (Pos. Feat.), absolute coordinates (Abs.) and reference grid (Ref.), are concatenated into input signals: (1) FNO (Li et al. (2021)) concatenates the input function value $a \in \mathbb{R}^{d_a}$ with the normalized absolute position coordinate $x \in \mathbb{R}^d$, usually scaled between 0 and 1. (2) Transolver (Wu et al. (2024a)), on the other hand, uses an advanced version which constructs a coarse reference grid and computes Euclidean distances from each input point to each point in this reference grid, concatenating these relative positions $x_{ref} \in \mathbb{R}^{(d_{ref})^d}$. In our work, we utilize the same absolute positional features as FNO to ensure a fair comparison.

- Since Transolver employs the multi-head in its implementation, it creates sub-grid with a equal size from the complete one.

- For Transolver, the optimizers and scheduler type is AdamW (Loshchilov & Hutter (2019)) and OneCycleLR, while for FNO and SKNO are Adam (Kingma & Lei Ba (2015)) and Cosine Annealing. All the learning rates are initially set to 0.001. More configurations can be found in the code provided in our supplementary materials.

**Zero-shot Super-resolution** To assess the resolution-invariant inference capability of SKNO, we evaluate the model on three benchmarks: 1D Burgers' equation (trained on a grid of size 128), 2D Darcy Flow (trained on a grid of size $85 \times 85$), and ERA5 Global Wind Field Prediction (trained on a grid of size $32 \times 32$). As shown in Tab. 10 and Tab. 12, SKNO maintains stability and accuracy even when tested on extremely fine grids, including 8192, $(421)^2$, and $(512)^2$. All experiments are conducted on a single RTX 4090 GPU, consistent with the setup used in other sections.

**Averaging Neural Operator on $D_p$** In experiments governed by non-linear equations, we utilize $F_p^{-1}\tilde{A}F_p + B$, where $\tilde{A}$ and $B$ are both matrices with diagonally enhanced initialization. Such kind of implementation can be thought as an Averaging Neural Operator (ANO) (Lanthaler et al. (2025)) along $p$, which governs the universal approximation of the PDE operator on $D_p$.

Table 7: Complete best training configurations

| Model | Epochs | Batch Size | # Modes *OR* Slices | # Layers | # Pos. Feat. / Type | # Sub-grid × size on $D_p$ |
|---|---|---|---|---|---|---|
| 1D Heat (Linear Block) | | | | | | |
| Transolver | 500 | 20 | 4 | 1 | N.A. | $1 \times 4$ |
| FNO | 500 | 20 | 1 | 1 | N.A. | $1 \times 4$ |
| **SKNO** | 500 | 20 | 1 | 1 | N.A. | $1 \times 4$ |
| 1D Advection (Linear Block) | | | | | | |
| Transolver | 500 | 4 | 8 | 1 | N.A. | $1 \times 16$ |
| FNO | 500 | 20 | 8 | 1 | N.A. | $1 \times 16$ |
| **SKNO** | 500 | 20 | 8 | 1 | N.A. | $1 \times 16$ |
| 1D Burgers | | | | | | |
| Transolver | 500 | 4 | 64 | 8 | 64 / Ref. | $8 \times 32$ |
| FNO | 500 | 20 | 16 | 4 | 1 / Abs. | $1 \times 64$ |
| **SKNO** | 500 | 20 | 16 | 4 | 1 / Abs. | $1 \times 64$ |
| 2D Gray-Scott | | | | | | |
| Transolver | 500 | 2 | 32 | 8 | $8^2 = 64$ / Ref. | $8 \times 32$ |
| FNO | 500 | 10 | 12 | 4 | 2 / Abs. | $1 \times 24$ |
| **SKNO** | 500 | 10 | 12 | 4 | 2 / Abs. | $1 \times 24$ |
| 2D Shallow-Water | | | | | | |
| Transolver | 100 | 2 | 32 | 8 | $8^2 = 64$ / Ref. | $8 \times 32$ |
| FNO | 100 | 20 | 8 | 4 | 2 / Abs. | $1 \times 24$ |
| **SKNO** | 100 | 20 | 8 | 4 | 2 / Abs. | $1 \times 24$ |
| 2D Darcy Flow | | | | | | |
| Transolver | 500 | 4 | 64 | 8 | $8^2 = 64$ / Ref. | $8 \times 16$ |
| FNO | 500 | 20 | 12 | 4 | 2 / Abs. | $1 \times 32$ |
| **SKNO** | 500 | 20 | 12 | 4+1 | 2 / Abs. | $1 \times 32$ |
| 2D Incompressible Navier-Stokes | | | | | | |
| Transolver | 500 | 2 | 32 | 8 | $8^2 = 64$ / Ref. | $8 \times 32$ |
| FNO | 8000 | 20 | 12 | 4 | 2 / Abs. | $1 \times 24$ |
| **SKNO** | 3000 | 20 | 12 | 4+1 | 2 / Abs. | $1 \times 24$ |
| 3D Compressible Navier-Stokes | | | | | | |
| Transolver | 100 | 2 | 32 | 8 | $8^3 = 512$ / Ref. | $8 \times 32$ |
| FNO | 100 | 10 | 16 | 4 | 3 / Abs. | $1 \times 32$ |
| **SKNO** | 100 | 10 | 16 | 4 | 3 / Abs. | $1 \times 32$ |
| 3D Rayleigh-Taylor Instability | | | | | | |
| Transolver | 250 | 2 | 32 | 8 | $8^3 = 512$ / Ref. | $8 \times 32$ |
| FNO | 250 | 10 | 16 | 4 | 3 / Abs. | $1 \times 32$ |
| **SKNO** | 250 | 10 | 16 | 4+1 | 3 / Abs. | $1 \times 32$ |
| ERA5 Wind Field Prediction | | | | | | |
| Transolver | 50 | 2 | 64 | 4 | $8^2 = 64$ / Ref. | $8 \times 8$ |
| FNO | 50 | 2 | 10 | 4 | 2 / Abs. | $1 \times 20$ |
| **SKNO** | 50 | 2 | 10 | 4 | 2 / Abs. | $1 \times 20$ |

## D  RELATED WORK

**Adding one dimension to solve PDEs**  Unlike traditional methods (Lucia et al. (2004); Chinesta et al. (2011)) that rely on dimension reduction techniques, such as symmetry analysis (Cantwell (2002)), to represent equations within their original domain, approaches like the level set method (Osher & Fedkiw (2001); Osher et al. (2004); Adalsteinsson & Sethian (1995); Jin & Osher (2003)) and the recent Schrödingerisation method (Jin & Liu (2024); Jin et al. (2023; 2024)) propose extending original PDEs to one more higher dimensional domains to create a pipe for functions to evolve. Similar "adding one dimension" ideas have also been used in signal processing like wavelets (Meyer (1992)) which employ scale and shift dimensions to represent original signals, and in quantum mechanics like the Wigner functions (Wigner (1932)) which utilize a single function displaying the probability (quasi-)distribution in the phase space.

**Neural Networks for PDEs**  In scenarios where PDEs are unknown or training data is limited, supervised learning methods are employed to approximate solution operators, which recently shows their capacity to universally represent solution operators for partial differential equations (PDEs) (Hornik et al. (1989); Kovachki et al. (2021); Lu et al. (2021); Lagaris et al. (1998)). Existing approaches mainly fall into two paradigms: Common Neural Networks and Neural Operators.

Common NNs approximate solution operators working on a fixed grid size, which maps discretized input to output signals using architectures such as MLPs (He et al. (2016); Chen et al. (2018)), CNNs (Qu et al. (2022); Brandstetter et al. (2023)). For example, (Obiols-Sales et al. (2020); Khoo et al.

(2021)) utilizes a typical CNN structure to map the input to the output image, which is composed of the solution of numerical solvers.

Neural operators (NOs) are designed to map between function spaces, enabling generalization across varying input discretizations without retraining (Kovachki et al. (2023); Wang et al. (2023)). One of the most popular NO definitions is the kernel integral operator (Li et al. (2021); Raonic et al. (2023b); Cao et al. (2024)). Early kernel function implementations employed graph-based message-passing networks (Li et al. (2020a;b)), where dense edge connections encoded these kernels. However, these methods faced scalability challenges due to long training times and inefficiencies in modeling interactions across multiple spatial and temporal scales. Transformer-based models (Kissas et al. (2022)), similar to graphs, also encounter limitations in computational efficiency due to quadratic time complexity. To address these challenges, tricky computation strategies on kernels have been proposed. Galerkin-based methods (Cao (2021); Ern & Guermond (2004)) employ linear projections to parameterize kernels, while (Xiao et al. (2024)) introduces a decomposition constrained by positive semi-definite assumptions. However, such methods suffer from the problems of reduced expressiveness due to the absence of non-linear activations like softmax. Recent SOTA model transolver (Wu et al. (2024a)) proposes a physics-attention module to approximate the kernel function with the aid of a self-attention-based aggregation on flattened data. While these graph or transformer-based advanced methods try to escape from the heavy quadratic complexity by tricky implementations on the kernel functions originally defined in (Li et al. (2020a; 2021)), they still pay the computational price for their neglect of the framework and model alignment in embedding spaces. While Mamba-based methods such as Tiwari et al. (2025) can further reduce the computational cost over the domain after aggregation, the representational and complexity bottlenecks in the embeddings remain unaddressed. Instead, empirically designed fully connected or multi-head architectures are typically retained by default to model the evolution along the auxiliary dimension.

Additionally, Fourier Neural Operators (FNO) (Li et al. (2021)) efficiently implement spectral kernel representations by global convolutions via the Fast Fourier Transform (FFT). While training fast, it also faces the representation bottleneck like other NO methods because of the unaligned evolution in embedding spaces. Subsequent works take efforts to extend the Fourier Transform to other analytically well-explored transformations. For example, Complex Neural Operator (Tiwari et al. (2023)) generalizes the Fourier Transformation with an extra learnable rotation parameter to model the $d$-dimensional kernel integration.

There is another NO definition: DeepONet (Lu et al. (2021)) employs a branch-trunk architecture inspired by the universal operator approximation theorem (Chen & Chen (1995)). These representations are combined through a dot product for the embeddings composed with functional outputs to predict the solution values. Recent advancements, such as integrating numerical solvers with DeepONet (Zhang et al. (2024)), have improved convergence by capturing multi-frequency components, enhancing performance on complex PDE systems. However, its general-purpose design may limit its effectiveness in scenarios requiring high accuracy and PDE-aligned structural representations.

## E  PROPERTIES OF SKNO

**Complexity Analysis**   To analyze computational complexity across general dimensional settings (1D, 2D, 3D, ...), let $N_x$ denote the number of grid points in the spatial domain, $N_p$ the grid size along the auxiliary dimension, and $k_x$ the number of aggregated slices/nodes (used in Transolver). The complexity comparison is summarized in Tab. 8.

Table 8: Complexity comparison of different kernel implementations.

| Models | Complexity |
|---|---|
| Transolver | $\mathcal{O}\big[N_p(N_x(k_x + N_p) + k_x^2)\big]$  $(k_x \geq 32,\ N_p = 128$ or $256)$ |
| FNO | $\mathcal{O}[N_p(N_x \log(N_x) + N_p k_x)]$  $(k_x \leq 16, N_p \leq 32)$ |
| **SKNO (Ours)** | $\mathcal{O}[N_p(N_x(\log N_x + \log N_p) + N_p k_x)]$  $(k_x \leq 16, N_p \leq 32)$ |

Consider an extreme but practically relevant case in high-accuracy numerical simulations: a 3D field with grid resolution $((2)^9)^3 = (512)^3$ and $N_p = 32$ (while Transolver typically requires $N_p = 128$

Table 9: Complexity comparison of different lifting implementations with positional encoding. (Assume all choose one linear layer)

| Models | Complexity |
|---|---|
| Transolver | $\mathcal{O}\big[N_p(d_a + (d_{ref})^d)\big]\ \ (d_{ref} = 8,\ N_p = 128 \text{ or } 256)$ |
| FNO | $\mathcal{O}[N_p(d_a + d)]\ \ (N_p \le 32)$ |
| **SKNO (Ours)** | $\mathcal{O}[N_p(d_a + d)]\ \ (N_p \le 32)$ |

for stable performance). Even with the minimal $k_x = 32$, Transolver's kernel implementation with a large linear constant ($k_x + N_p > 128$) leads to a computational cost much higher than SKNO ($\log N_x = 27$) or FNO ($\log N_x N_p = 27 + 5 = 32$).

In lower dimensions like 1D or 2D cases, where $N_x$ is exponentially smaller, our kernel retains the linear-complexity advantages while achieving superior performance with significantly smaller scale constants compared to the SOTA Transolver (under the log benefit from Fast Fourier Transform), which we have validated across various benchmarks (see Tab.3, Tab.11 and 13). It is worth noting that, when the positional encodings are removed in Table 9, the prediction error of Transolver increases dramatically (from $6e-3$ to $6e-1$ in relative $L_2$ error on Burgers), whereas the other two models maintain nearly the same accuracy level.

**Spectral Interpretation**  Since our SKNO adopts a spectral implementation, we primarily compare it to the FNO (Li et al. (2021)) as a reference framework. This choice also facilitates generalization to other spectral models like (Guibas et al. (2022); Li et al. (2024a)). To avoid notational clutter, we begin with the $d = 1$ case, noting that the following arguments extend straightforwardly to higher dimensions.

**Theorem 1** (Neural Operator along the Auxiliary Dimension). *Let $v(x, p)$ be taken from a compact set $V$ in the appropriate Sobolev space so that its truncated Fourier representatives*

$$\hat{\boldsymbol{v}}_i := F_{k_x}^x[P_{k_x}^x v](\hat{\boldsymbol{x}}_i, \boldsymbol{p}) \in \hat{V}_i \subset \mathbb{C}^{N_p}, \qquad \tilde{\hat{\boldsymbol{v}}}_i := F^p[R_{k_p}^p(F_{k_x}^x[P_{k_x}^x v])(\hat{\boldsymbol{x}}_i, \tilde{\boldsymbol{p}})) \in \tilde{\hat{V}}_i \subset \mathbb{C}^{N_p},$$

*are compact sets in a finite coefficient space. Then*

1. *Approximating an operator $\mathcal{M}$ is equivalent (via the Fourier-conjugate decomposition) to approximating the conjugate mapping(s) $\{\widehat{\mathcal{M}}_i\}_{i=1}^{k_x}$ (and, after also Fourier-transforming in $p$, the conjugates $\{\widetilde{\mathcal{M}}_i\}$).*

2. *The forward DFT/frequency-projection and the inverse discrete transform admit neural operator approximation on the spectral domain (See Lemma 7 and Lemma 8 in (Kovachki et al. (2021))), hence the operator learning problem reduces to a finite-dimensional neural-network approximation on the compact coefficient sets $\hat{V}_i$, $\tilde{\hat{V}}_i$.*

*In particular, treating the auxiliary $p$-dimension via an additional DFT (as SKNO does) simply produces an enlarged conjugate mapping; the same approximation strategy (as Lemma 7 + Lemma 8 + finite NN on compact coefficients) applies.*

*Note. $\widehat{\mathcal{M}}_i : \hat{V}_i \subset \mathbb{R}^{2N_p} \to \mathbb{R}^{2N_p}$ for $i = 1, ..., k_x$ and $\widetilde{\mathcal{M}}_i : \tilde{\hat{V}}_i \subset \mathbb{R}^{2N_p} \to \mathbb{R}^{2N_p}$. Usually, we take $k_p = N_p$ and the $\frac{length(\boldsymbol{p}) - k_p}{length(\boldsymbol{p})}$ random dropout $R_{k_p}^p = I^p$.*

*Proof.* By the Fourier-conjugate decomposition (Eq. (12) in (Kovachki et al. (2021))), the operator can be written as

$$\begin{aligned}\mathcal{M} &= (F_{k_x}^x)^{-1} \circ \widehat{\mathcal{M}}_{k_x} \circ F_{k_x}^x \circ P_{k_x}^x \\ &= (F_{k_x}^x)^{-1} \circ (F^p)^{-1} \circ \widetilde{\mathcal{M}}_{k_x} \circ F^p \circ F_{k_x}^x \circ P_{k_x}^x.\end{aligned} \tag{33}$$

Here, the second line follows from additionally applying a DFT in the auxiliary variable $p$.

The mapping in Equ.33 $F^p F^x_{k_x} [P^x_{k_x}(\cdot)]$ sends the compact input family $V$ into a compact subset of Fourier coefficient space. Lemma 7 and Lemma 8 of (Kovachki et al. (2021)) guarantee that the left-hand and right-hand parts of both $\widehat{\mathcal{M}}_{k_x}$ and $\widetilde{\mathcal{M}}_{k_x}$ can be approximated within the desired accuracy.

Thus, the problem reduces to approximating the conjugate operators $\widehat{\mathcal{M}}_{k_x}$ or $\widetilde{\mathcal{M}}_{k_x}$, both of which are continuous maps defined on compact subsets of $\mathbb{R}^{2N_p}$. By the universal approximation theorem for neural networks (Barron (2002); Hornik et al. (1989)), any such continuous map can be approximated to arbitrary accuracy by a finite neural network. Hence, there exists finite NNs that approximate the conjugate mappings on $\hat{V}_i$ (or $\tilde{\hat{V}}_i$) within the desired accuracy.

Easy to show that SKNO universally approximates any $d$-dimensional operator, see Theorem 5 in (Kovachki et al. (2021)).

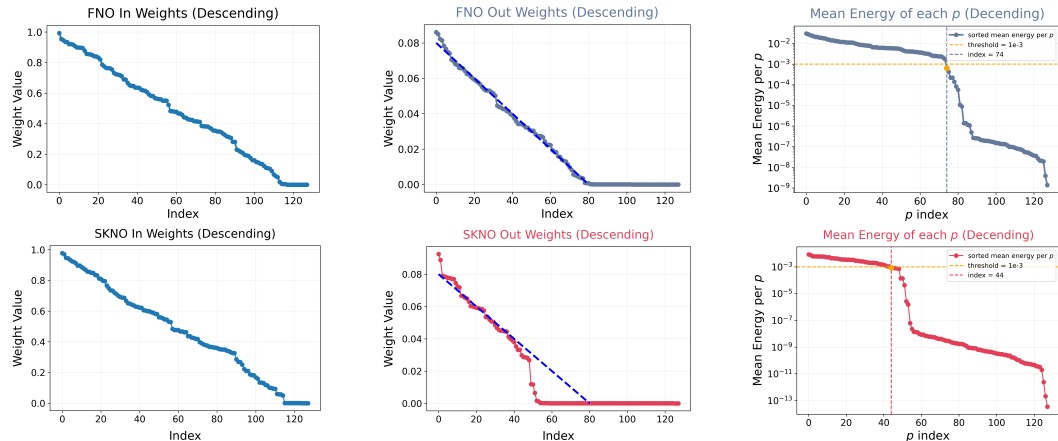

Figure 6: Dictionary comparison of FNO and SKNO. We train both models with a sufficiently large recovering budget $N_p$ on the Burgers benchmark and visualize the absolute values of the input/output weights, together with the mean energy of the dictionary of pattern fields. While the input weights of FNO (top left) and SKNO (bottom left) exhibit nearly identical distributions, SKNO yields a noticeably sparser structure in its output weights (bottom middle) and in the mean energy of each pattern field (bottom right) compared to FNO. The yellow horizontal line denotes a mean-energy threshold of $10^{-3}$, where modes below this threshold contribute only marginally to the final prediction. For a complete visualization of the output-weight and pattern-field pairs in the dictionary, see Fig. 16.

**Sparse Representation** Due to the limited network width, $N_p$ should be finite and as small as possible. Benefiting from the PDE-aligned propagator design, SKNO constructs a sparser dictionary that is more efficient in capturing the energy of the solutions than FNO, which will be discussed below with visualizations and formal proofs.

For a clear and fair comparison, we consider SKNO without the differential operator and the baseline FNO. The lifting and recovering operators in both models are chosen as unbiased linear layers without positional encoding. Under the same configurations, we compare the dictionaries of SKNO and FNO at the last output linear layer in Fig.6, which illustrates the sparsity advantage of the SKNO dictionary.

Beyond the experiments, we also provide a formal proof in a finite-dimensional matrix–vector setting, which aligns well with the neural implementation, to clarify the efficient energy capture of SKNO. The extension to function spaces amounts to working in the corresponding infinite-dimensional setting.

**Lemma 1** (Preserved Energy after Top-$r$ Truncation). *For a general matrix–vector multiplication $\boldsymbol{u} = G\boldsymbol{y}$ where $G \in \mathbb{R}^{N_x \times N_p}$ and $N_x \geq N_p$, if we perform the top-$r$ approximation $G_r$ using the*

*singular value decomposition (SVD) of G, the energy preserved in the prediction $G_r \boldsymbol{y}$ is*

$$\|G_r \boldsymbol{y}\|_2^2 = \sum_{m=1}^{r} \lambda_m^2 \langle \boldsymbol{y}, \phi_m \rangle^2, \quad r \leq N_x, \tag{34}$$

*where $G_r \boldsymbol{y} = \Psi_r \Lambda_r \Phi_r^T \boldsymbol{y} = \sum_{m=1}^{r} \lambda_m \langle \boldsymbol{y}, \phi_m \rangle \psi_m$, and $\psi_m$, $\phi_m$, and $\lambda_m$ are the m-th left singular vector, right singular vector, and singular value, respectively.*

*Proof.* This follows directly from $G_r \boldsymbol{y}$ representing with the orthonormal basis $\{\psi_m\}$ of range($r$).

**Theorem 2** (Efficient Energy Capture with Limited Width $r < N_p$). *Assuming the models have the same $\mathcal{P}$, we have*

$$\left\| u(x) - \sum_{m=1}^{r} \chi_m^{(\text{FNO})} (v_L)_m^{(\text{FNO})}(x) \right\|_2 - \left\| u(x) - \sum_{m=1}^{r} \chi_m^{(\text{SKNO})} (v_L)_m^{(\text{SKNO})}(x) \right\|_2 = \|u(x)\|_2 \cdot \epsilon(r),$$
$$\tag{35}$$

*where $\epsilon(r)$ is a non-increasing function with respect to $r$. When $r \to N_p$, $\epsilon(r) \to 0$.*

*Note.* (1) The length $N_x$ in the following discussion is obtained by flattening the $d$-dimensional tensor while keeping the sequential order. (2) Without loss of generality, we discuss the case $d_u = 1$.

*Proof.* Denote the vector $\boldsymbol{u} \in \mathbb{R}^{N_x}$ discretized on $D_x$, the vector $\boldsymbol{\chi}^{(\text{M})} \in \mathbb{R}^{N_p}$ discretized on $N_p$, the matrix $V_L^{(\text{M})} \in \mathbb{R}^{N_x \times N_p}$ where $N_x \geq N_p$. We focus on the last output (linear) layer in neural operators M. If $N_p$ is sufficiently large, we have

$$\boldsymbol{u} = V_L^{(\text{M})} \boldsymbol{\chi}^{(\text{M})}$$
$$= \Psi_L^{(\text{M})} \Lambda_L^{(\text{M})} (\Phi_L^{(\text{M})})^T \boldsymbol{\chi}^{(\text{M})}, \tag{36}$$

with singular values $\lambda_1^{(\text{M})} \geq \lambda_2^{(\text{M})} \geq \cdots \geq \lambda_{N_x}^{(\text{M})}$. The matrices $\Psi_L^{(\text{M})} = (\psi_1^{(\text{M})}, \psi_2^{(\text{M})}, \ldots, \psi_{N_x}^{(\text{M})})$ and $\Phi_L^{(\text{M})} = (\phi_1^{(\text{M})}, \phi_2^{(\text{M})}, \ldots, \phi_{N_p}^{(\text{M})})$ are composed with their column vectors $\{\psi_m\}_{m=1}^{N_x}$ and $\{\phi_m\}_{m=1}^{N_p}$, respectively.

In practice, we apply the $r$-truncation on $N_p$ as the optimal approximation under the metric in Equ. 16. Utilizing the rank-$r$ truncated SVD, we have

$$(V_L^{(\text{M})})_r := (\Psi_L^{(\text{M})})_r (\Lambda_L^{(\text{M})})_r (\Phi_L^{(\text{M})})_r^T$$
$$= \sum_{m=1}^{r} \lambda_m^{(\text{M})} \psi_m^{(\text{M})} (\phi_m^{(\text{M})})^T, \ r \leq N_x. \tag{37}$$

From Lemma 1, the (relative) energy of $u$ captured by the top-$r$ SVD modes of model M is

$$E^{(\text{M})}(r) := \frac{\|(V_L^{(\text{M})})_r \boldsymbol{\chi}^{(\text{M})}\|_2^2}{\|\boldsymbol{u}\|_2^2}$$
$$= \frac{\sum_{m=1}^{r} (\lambda_m^{(\text{M})})^2 \langle \boldsymbol{\chi}^{(\text{M})}, \phi_m^{(\text{M})} \rangle^2}{\sum_{m=1}^{N_x} (\lambda_m^{(\text{M})})^2 \langle \boldsymbol{\chi}^{(\text{M})}, \phi_m^{(\text{M})} \rangle^2}. \tag{38}$$

Since singular values are non-negative, the sequence $E^{(\text{M})}(r)$ is non-decreasing with respect to $r$ for each model M. And the truncation error is

$$\|\boldsymbol{u} - (V_L^{(\text{M})})_r \boldsymbol{\chi}^{(\text{M})}\|_2 = \|\boldsymbol{u}\|_2 \sqrt{1 - E^{(\text{M})}(r)}, \quad r = 1, \ldots, N_x. \tag{39}$$

Specifically, the error difference between FNO and SKNO is

$$\epsilon(r) = \frac{\|\boldsymbol{u} - (V_L^{(\text{FNO})})_r \boldsymbol{\chi}\|_2 - \|\boldsymbol{u} - (V_L^{(\text{SKNO})})_r \boldsymbol{\chi}\|_2}{\|\boldsymbol{u}\|_2}$$

$$:= \sqrt{1 - E^{(\text{FNO})}(r)} - \sqrt{1 - E^{(\text{SKNO})}(r)}. \tag{40}$$

From Equ.38, $\epsilon(r) \to 0$ as $r \to N_p$.

Let $\mathcal{H}^{(\text{M})}$ denotes the set of $V_L^{(\text{M})}$ that can be realized by NO M under the same width budget $r$. We have

$$E^{(\text{M})}(r) := \sup_{V_L^{(\text{M})} \in \mathcal{H}_{\text{M}}} \frac{\left\|(V_L^{(\text{M})})_r \boldsymbol{\chi}^{(\text{M})}\right\|_2^2}{\|\boldsymbol{u}\|_2^2}, \qquad \text{M} \in \{\text{FNO}, \text{SKNO}\}, \ r \le N_p. \tag{41}$$

Since $\mathcal{H}_{\text{FNO}} \subset \mathcal{H}_{\text{SKNO}}$, the supremum over $\mathcal{H}_{\text{SKNO}}$ cannot be smaller than the supremum over $\mathcal{H}_{\text{FNO}}$, which implies

$$E^{(\text{SKNO})}(r) \ge E^{(\text{FNO})}(r), \ \forall \, r \le N_p. \tag{42}$$

Thus, for all $r = 1, \ldots, N_p$

$$\epsilon(r) \ge 0, \tag{43}$$

It is straightforward to extend the above arguments to function spaces.

**Kernel Function Dependency**

**Theorem 3** (Relationship between $d$- and $d+1$-Dimensional Kernel Integral Operators). *Let $\mathcal{G}$ be a $d$-dimensional kernel integral operator with kernel $\kappa_d(x, y)$, and let $\mathcal{M} := \mathcal{K}$ denote a $(d+1)$-dimensional operator with kernel*

$$\kappa_{d+1}(x, y, p, p') = \kappa(x, y, p, p')\chi(p)w^T(p').$$

*Then,*

$$\kappa_d(x, y) = \iint_{D_p \times D_p} \kappa_{d+1}(x, y, p, p') \, dp' \, dp, \tag{44}$$

*where both $\kappa_d$ and $\kappa_{d+1}$ are matrices of size $d_u \times d_a$.*

*Proof.* Starting from Equ. 7–10 and letting $\mathcal{M} = \mathcal{K}$, we obtain

$$u(x) = \int_{D_p} \chi(p)\mathcal{M}[w^T a](x, p) \, dp$$

$$= \int_{D_p} \chi(p) \iint_{D_x \times D_p} \kappa(x, y, p, p')w^T(p')a(y) \, dy \, dp' \, dp$$

$$= \int_{D_p} \iint_{D_p \times D_x} \kappa(x, y, p, p')\chi(p)w^T(p')a(y) \, dy \, dp' \, dp. \tag{45}$$

Defining $\kappa_{d+1}(x, y, p, p') := \kappa(x, y, p, p')\chi(p)w^T(p')$, Equ. 45 becomes

$$u(x) = \int_{D_x} \left( \iint_{D_p \times D_p} \kappa_{d+1}(x, y, p, p') \, dp' \, dp \right) a(y) \, dy. \tag{46}$$

By definition, the $d$-dimensional operator satisfies

$$\mathcal{G}[a](x) = \int_{D_x} \kappa_d(x, y)a(y) \, dy, \tag{47}$$

which implies

$$\kappa_d(x,y) = \iint_{D_p \times D_p} \kappa_{d+1}(x,y,p,p') \, dp' \, dp. \tag{48}$$

Hence, the $d$-dimensional kernel function $\kappa_d$ can be physically interpreted as the projection of the $(d+1)$-dimensional kernel $\kappa_{d+1}$ over the auxiliary variable $p$ in the constructed phase space. Moreover, from Eq. 46, different choices of $w(p)$ and $\chi(p)$ directly influence the function form of $\kappa$ and thereby affect the approximation accuracy under the same model structure, as shown in Sec. 4.2.

**Adjoint Backward Propagation**    Under the squared loss function like Equ. 16, we show the adjoint backward propagation property of SKNO.

**Theorem 4** (Backward Residual Propagation). *Here we show the backward updates around linear kernel integrations in* SKNO. *Assume $d_a = d_u = 1$ and denote the matrix multiplication on the last output layer $V_L^{(\text{SKNO})} \boldsymbol{\chi}^{(\text{SKNO})}$ and the backward input residual $\boldsymbol{r}^{(\text{SKNO})}$, where the vector $\boldsymbol{\chi}^{(\text{SKNO})} \in \mathbb{R}^{N_p}$ is discretized on $N_p$, and the matrix $V_L^{(\text{SKNO})} \in \mathbb{R}^{N_x \times N_p}$ where $N_x > N_p$. The backpropagation (Rumelhart et al. (1986)) update*

$$\Delta \boldsymbol{\chi}^{(\text{SKNO})} = \sum_i (V_L^{(\text{SKNO})})_{i\cdot}^T \cdot \boldsymbol{r}_i^{(\text{SKNO})} \tag{49}$$

$$\text{Grad}(V_L^{(\text{SKNO})}) = \boldsymbol{r}^{(\text{SKNO})} \cdot (\boldsymbol{\chi}^{(\text{SKNO})})^T \tag{50}$$

$$FC : (\Delta \widehat{\mathbf{B}})_{ijj'} = \text{conj}(\widehat{V}_{L-1})_{ij} \cdot \hat{\boldsymbol{r}}_i^{(\text{SKNO})} \cdot \mathbf{M}_{ij'}^{(\text{SKNO})} \cdot (\boldsymbol{\chi}^{(\text{SKNO})})_{j'}^T \tag{51}$$

$$\text{Grad}(\widehat{V}_{L-1}^{(\text{SKNO})})_{ij} = \sum_{j'} \widehat{\mathbf{B}}_{ijj'} \cdot \hat{\boldsymbol{r}}_i^{(\text{SKNO})} \cdot \mathbf{M}_{ij'}^{(\text{SKNO})} \cdot (\boldsymbol{\chi}^{(\text{SKNO})})_{j'}^T \tag{52}$$

$$(\Delta \widehat{\widetilde{\mathbf{A}}})_{ijj'} = \text{conj}(\widehat{\widetilde{V}}_{L-1})_{ij} \cdot \hat{\boldsymbol{r}}_i^{(\text{SKNO})} \cdot \mathbf{M}_{ij'}^{(\text{SKNO})} \cdot (\tilde{\boldsymbol{\chi}}^{(\text{SKNO})})_{j'}^T \tag{53}$$

$$\text{Grad}(\widehat{\widetilde{V}}_{L-1}^{(\text{SKNO})})_{ij} = \sum_{j'} \widehat{\widetilde{\mathbf{A}}}_{ijj'} \cdot \hat{\boldsymbol{r}}_i^{(\text{SKNO})} \cdot \mathbf{M}_{ij'}^{(\text{SKNO})} \cdot (\tilde{\boldsymbol{\chi}}^{(\text{SKNO})})_{j'}^T \tag{54}$$

$$Diag : (\Delta \widehat{\mathbf{B}})_{ij} = \text{conj}(\widehat{V}_{L-1})_{ij} \cdot \hat{\boldsymbol{r}}_i^{(\text{SKNO})} \cdot \mathbf{M}_{ij}^{(\text{SKNO})} \cdot (\boldsymbol{\chi}^{(\text{SKNO})})_j^T \tag{55}$$

$$\text{Grad}(\widehat{V}_{L-1}^{(\text{SKNO})})_{ij} = \widehat{\mathbf{B}}_{ij} \cdot \hat{\boldsymbol{r}}_i^{(\text{SKNO})} \cdot \mathbf{M}_{ij}^{(\text{SKNO})} \cdot (\boldsymbol{\chi}^{(\text{SKNO})})_j^T \tag{56}$$

$$(\Delta \widehat{\widetilde{\mathbf{A}}})_{ij} = \text{conj}(\widehat{\widetilde{V}}_{L-1})_{ij} \cdot \hat{\boldsymbol{r}}_i^{(\text{SKNO})} \cdot \mathbf{M}_{ij}^{(\text{SKNO})} \cdot (\tilde{\boldsymbol{\chi}}^{(\text{SKNO})})_j^T \tag{57}$$

$$\text{Grad}((\widehat{\widetilde{V}}_{L-1}^{(\text{SKNO})})_{ij} = \widehat{\widetilde{\mathbf{A}}}_{ij} \cdot \hat{\boldsymbol{r}}_i^{(\text{SKNO})} \cdot \mathbf{M}_{ij}^{(\text{SKNO})} \cdot (\tilde{\boldsymbol{\chi}}^{(\text{SKNO})})_j^T, \tag{58}$$

*where the weight $\Delta \boldsymbol{\chi} \in \mathbb{R}^{N_p}$, complex tensors $\Delta \widehat{\mathbf{B}} \in \mathbb{C}^{N_x \times N_p \times N_p}$ or $\mathbb{C}^{N_x \times N_p}$ and $\Delta \widehat{\widetilde{\mathbf{A}}} \in \mathbb{C}^{N_x \times N_p \times N_p}$ or $\mathbb{C}^{N_x \times N_p}$ are the changing values of parameters.* $\text{conj}(\cdot)$ *denotes element-wise taking the conjugate value of the input.* $\mathbf{M}_{ij}^{(\text{SKNO})} = \mathbf{1}[(V_L'^{(\text{SKNO})})_{ij} > 0]$ *where $V_L^{(\text{SKNO})} = \text{ReLU}(V_L'^{(\text{SKNO})})$.*

*Proof.* Easy to derive these updating values by taking partial derivatives with respect to the parameters in propagators $\widehat{\mathbf{B}}$ and $\widehat{\widetilde{\mathbf{A}}}$ and in the recovering operator $\boldsymbol{\chi}$. A notable point here is the conjugate gradient backpropagation path to the Fourier domain like the $\hat{\boldsymbol{r}}_i^{(\text{SKNO})}$ like in Equ.52.

The adjoint backward residual propagation and conjugate parameter updates shown above are directly derived from SKNO's spectral implementation. From Remark 4, we can generalize the backward propagation framework for $d+1$-dimensional neural operators as follows

$$\Delta v_L(x,p) = \mathcal{Q}^*[r](x,p) = \chi^T(p) r(x) \tag{59}$$

$$\mathcal{K}^*[\Delta v](x,p) = \mathcal{K}[\Delta v](x,p) = \iint_{D_x \times D_p} \kappa(x,y,p,p') \Delta v(y,p') dy dp' \tag{60}$$

$$\Delta a(x) = \mathcal{P}^*[\Delta v_0](x) = \int_{D_p} \omega(p) \Delta v_0(x,p) dp, \tag{61}$$

where $r(x) := \Delta u(x) = u(x) - \mathcal{G}[a](x)$ denotes the residual between the solution $u(x)$ and the corresponding forward prediction $\mathcal{G}[a](x)$, and we define the adjoint operator of the lifting $\mathcal{P}^*$ to have a $p$-integral form as the recovering operator $\mathcal{Q}$ in Eq. 9, and vice versa. See Lemma 2.

For previous $d$ dimensional kernel integration designs, an additional matrix inversion is required when formulating the backpropagation process, whereas $d+1$ dimensional neural operators are allowed to share the same expression for the forward and adjoint operators, i.e., $\mathcal{K} = \mathcal{K}^*$. When $d+1$ dimensional neural operators converge, the backpropagation updates (Rumelhart et al. (1986)) stabilize within a PDE-specific equivalence class of parameters, up to a certain error tolerance.

**Lemma 2** (Adjoint of the recovering operator $\mathcal{Q}$). *Let $H_x := L^2(D_x)$ and $H_{x,p} := L^2(D_x \times D_p)$ with the standard $L^2$ inner products. Then $\mathcal{Q}^*$ is the adjoint of $\mathcal{Q}$, i.e.*

$$\langle \mathcal{Q}[v], r \rangle_{H_x} = \langle v, \mathcal{Q}^*[r] \rangle_{H_{x,p}} \quad \forall v \in H_{x,p},\ r \in H_x.$$

*Proof.* By definition of the $L^2$ inner product on $H_x$,

$$\begin{aligned}
\langle \mathcal{Q}[v], r \rangle_{H_x} &= \int_{D_x} (\mathcal{Q}[v])(x)\, r(x)\, dx \\
&= \int_{D_x} \left( \int_{D_p} \chi(p)\, v(x,p)\, dp \right) r(x)\, dx \\
&= \int_{D_x} \int_{D_p} v(x,p)\, \chi(p)\, r(x)\, dp\, dx.
\end{aligned} \tag{62}$$

By definition of the $L^2$ inner product on $H_{x,p}$,

$$\begin{aligned}
\langle v, \mathcal{Q}^*[r] \rangle_{H_{x,p}} &= \int_{D_x} \int_{D_p} v(x,p)\, (\mathcal{Q}^*[r])(x,p)\, dp\, dx \\
&= \int_{D_x} \int_{D_p} v(x,p)\, \chi(p)\, r(x)\, dp\, dx.
\end{aligned} \tag{63}$$

Thus $\langle \mathcal{Q}[v], r \rangle_{H_x} = \langle v, \mathcal{Q}^*[r] \rangle_{H_{x,p}}$ for all $v, r$, so $\mathcal{Q}^*$ is the adjoint of $\mathcal{Q}$. Similar proof for the adjoint between $\mathcal{P}$ and $\mathcal{P}^*$.

**Entangled Systems** Moreover, we novelly compute the entanglement entropy values after each $d+1$ dimensional propagation, inspired by the quantum view in Remark 2, and plot some randomly selected cases of them in Fig.7: Compared to FNO, SKNO introduces stronger perturbations on the singular spectrum after $\mathcal{L}_l[v_l]$, cooperated with effectively controlling the sparsity of the energy distribution by $\sigma_l$. In addition, the PDE-aligned design in the $\mathcal{L}_l[v_l]$ of SKNO makes it more flexible to reduce entanglement in the final linear stage further, while FNO is purely driven by the squared-loss objective to substantially perturb the $(d+1)$-dimensional signal without an implicit structural regularization along $p$, which leads to increased entanglement entropy in some cases.

**Remark 2** (Propagation of the $d+1$ Entangled States). *From (Jin et al. (2022)), if $\mathcal{P}$ is chosen as a linear layer, $v_0(x,p)$ is a pure global quantum state without entanglement between the original $x$ and $p$ spaces. In neural operators, the initial quantum state $V_0$ becomes entangled during the (non-linear) forward propagation. A measure of the degree of such entanglement between two subsystems of a composite quantum system is the entanglement entropy, which can be calculated numerically using Algorithm 1.*

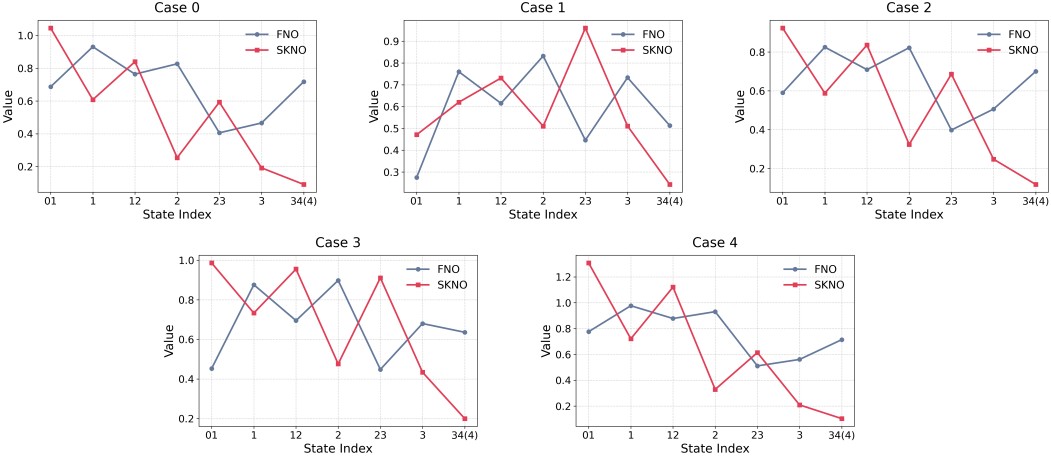

Figure 7: Entanglement entropy of intermediate layer outputs. For five randomly selected test cases from the test set, the final entanglement entropy of SKNO is consistently lower than that of FNO, indicating that the $(d+1)$-dimensional design of SKNO yields representations that are less entangled and exhibit a more energy-concentrated distribution. On the horizontal axis, a two-digit index (e.g., "01") denotes the output after the $\mathcal{L}_0$ and before the $\sigma_0$ between $v_0$ and $v_1$, whereas a single-digit index (e.g., "1") denotes $v_1$ as the output of the first propagator ($\sigma_0 \circ \mathcal{L}_0$). Note that the initial index "0" corresponds to $v_0$. Under a purely linear lifting operator without positional encoding, $v_0$ is typically separable, so all curves effectively start from an unentangled state at the origin and evolve through the forward propagation.

---

**Algorithm 1** Entanglement Entropy Numerical Calculation

---

**Require:** $V_l$, $l = 0, 1, ..., L$.
**Ensure:** Entanglement Entropy $S_l$.
  **Perform singular value decomposition:**

$$V_l = \Psi_l \Lambda_l (\Phi_l)^T$$

  where $\Sigma_l = \mathrm{diag}((\lambda_1)_l, (\lambda_2)_l, \ldots, (\lambda_{N_p})_l)$.
  **Normalize Schmidt coefficients:**

$$(c_m)_l \leftarrow \frac{(\lambda_m)_l}{\sqrt{\sum_{m'}(\lambda_{m'})_l^2}}.$$

  **Compute entanglement entropy:**

$$S_l \leftarrow -\sum_m (c_m)_l^2 \log(c_m)_l^2.$$

---

## F VISUALIZATION

**Patterns with different scales** We visualize cases to highlight the patterns with different scales of our SKNO. These subplots are plotted over all $p$ values after the $d + 1$ dimensional evolution. Cases include the 1D Burgers, 2D Darcy Flow, and 2D incompressible Navier-Stokes benchmarks, shown in Fig. 9, Fig. 10 and Fig. 12 respectively.

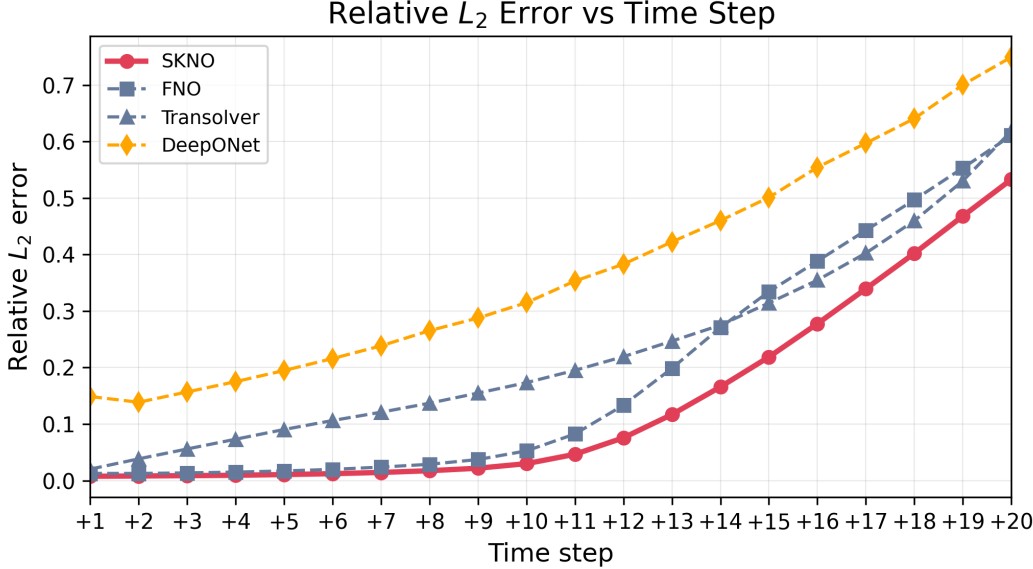

Figure 8: Visualization of zero-shot predictions at future time steps. For the 2D incompressible Navier–Stokes with viscosity $1 \times 10^{-4}$ Li et al. (2021), all models are autoregressively evaluated over the twenty time steps following the input time. SKNO maintains its state-of-the-art performance on both the seen temporal range (+1 to +10) and the unseen temporal range (+11 to +20), achieving a substantial performance margin over competing methods. Compared with FNO, Transolver, and DeepONet, SKNO achieves average error reductions of 34.5%, 58.7%, and 74.5%, respectively, across all shown predictions. Note that even on the first predicted step, SKNO attains an error of 0.0076, outperforming FNO (0.0127), Transolver (0.0202), and DeepONet (0.1488).

Table 10: Prediction errors of neural operators on the 1D Burgers equation and 2D Darcy flow across different grid sizes. The table also reports the variance of errors over grid resolutions, highlighting SKNO's superior accuracy and stability.

| | | | *1D Burgers Equation* | | | | | |
|---|---|---|---|---|---|---|---|---|
| **Grid Size** | 128 | 256 | 512 | 1024 | 2048 | 4096 | 8192 | **Variance** |
| Transolver | 6.277E-03 | 7.186E-03 | 8.058E-03 | 8.546E-03 | 8.800E-03 | 8.928E-03 | 8.993E-03 | 9.04472E-07 |
| FNO | 6.479E-04 | 6.476E-04 | 6.480E-04 | 6.477E-04 | 6.454E-04 | 6.455E-04 | 6.455E-04 | 1.33912E-12 |
| **SKNO** | **5.475E-04** | **5.489E-04** | **5.490E-04** | **5.489E-04** | **5.488E-04** | **5.487E-04** | **5.487E-04** | **2.53997E-13** |
| | | | *2D Darcy Flow* | | | | | |
| **Grid Size** | $(22)^2$ | $(29)^2$ | $(43)^2$ | $(85)^2$ | $(141)^2$ | $(211)^2$ | $(421)^2$ | **Variance** |
| FNO | 9.813E-02 | 7.718E-02 | 4.956E-02 | 6.165E-03 | 3.044E-02 | 3.897E-02 | 4.939E-02 | 7.83082E-04 |
| **SKNO** | **7.739E-02** | **6.302E-02** | **4.122E-02** | **5.551E-03** | **2.366E-02** | **3.271E-02** | **4.112E-02** | **4.90633E-04** |

Table 11: Performance of neural operator models on 2D Darcy Flow, 2D Shallow Water and 3D Compressible NS benchmarks. On the 2D Shallow Water, we train models on a randomly selected resolution and evaluate them at the highest resolution. –: No 3D implementation available; *: Requires predefining the resolution, here trained on the highest resolution; **: For fairness, training times are all reported on the highest resolutions. The CNO model is described in (Raonic et al. (2023a)).

| **Models** | DeepONet | CNO | FNO | Transolver | SKNO |
|---|---|---|---|---|---|
| | *2D Darcy Flow* | | | | |
| Rel. $L_2$ Err. | 0.0609 | 0.0060 | 0.0062 | 0.0058 | **0.0055** |
| Training Time | **5.20 min** | 63.34 min | 10.21 min | 115.48 min | 17.59 min |
| | *2D Shallow Water* | | | | |
| Rel. $L_2$ Err. | 0.0902* | 0.0085* | 0.0030 | 0.0088 | **0.0027** |
| Training Time** | **8.59 min** | 49.26 min | 11.43 min | 128.64 min | 20.85 min |
| | *3D Compressible Navier-Stokes* | | | | |
| Rel. $L_2$ Err. | 0.5628 | – | 0.2631 | 0.2947 | **0.2376** |
| Training Time | **3.90 min** | – | 10.12 min | 361.41 min | 23.33 min |

Table 12: Zero-shot super-resolution evaluation on ERA5 wind field prediction. As the grid resolution increases, SKNO exhibits substantially lower growth in relative $L_2$ error compared to FNO, leading to a pronounced accuracy advantage on the finest grid $(512)^2$.

| | *Wind U* | | | | | |
|---|---|---|---|---|---|---|
| **Grid Size** | $(32)^2$ | $(64)^2$ | $(128)^2$ | $(256)^2$ | $(512)^2$ | **Variance** |
| FNO | 6.272E-02 | 6.525E-02 | 6.872E-02 | 7.097E-02 | 7.212E-02 | 1.23506E-05 |
| **SKNO** | **6.164E-02** | **6.237E-02** | **6.366E-02** | **6.445E-02** | **6.495E-02** | **1.54763E-06** |
| | *Wind V* | | | | | |
| **Grid Size** | $(32)^2$ | $(64)^2$ | $(128)^2$ | $(256)^2$ | $(512)^2$ | **Variance** |
| FNO | 2.165E-01 | 2.326E-01 | 2.430E-01 | 2.476E-01 | 2.498E-01 | 1.49635E-04 |
| **SKNO** | **1.969E-01** | **2.027E-01** | **2.108E-01** | **2.156E-01** | **2.181E-01** | **6.30471E-05** |

Table 13: Results for ERA5 wind field prediction.

| Performance / Model | Training Time | Wind U Error | Wind V Error |
|---|---|---|---|
| FNO | **0.76 min** | 7.064e-2 | 2.133e-1 |
| Transolver | 10.81 min | 6.291e-2 | 2.007e-1 |
| SKNO | 1.75 min | **6.111e-2** | **1.920e-1** |

Table 14: Performance of models on 2D stress and 2D strain benchmarks (Rashid et al. (2022); Burark et al. (2024)). The reported average (Avg.) training time is the mean over these two benchmarks, with all other configurations identical except for the type of predicted field.

| **Models** | DeepONet | FNO | Transolver | SKNO |
|---|---|---|---|---|
| | *2D Stress* | | | |
| Rel. $L_2$ Err. | 0.3891 | 0.0622 | 0.0481 | **0.0472** |
| | *2D Strain* | | | |
| Rel. $L_2$ Err. | 0.5439 | 0.0575 | 0.0465 | **0.0463** |
| Avg. Training Time | **4.15 min** | 9.43 min | 117.39 min | 19.36 min |

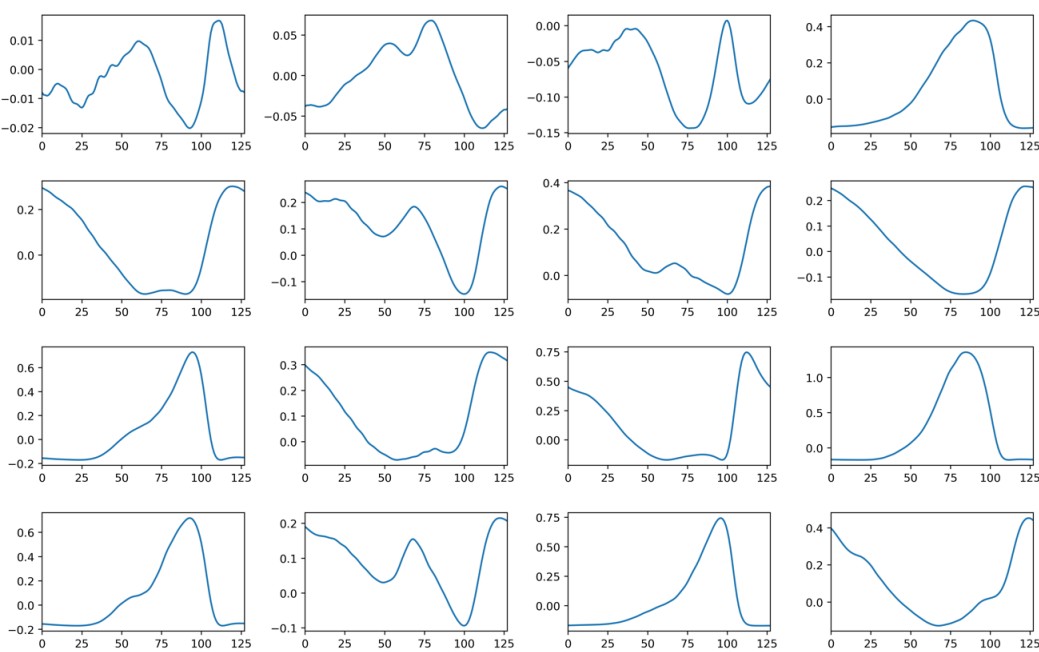

Figure 9: Pattern visualization of the 1D Burgers case.

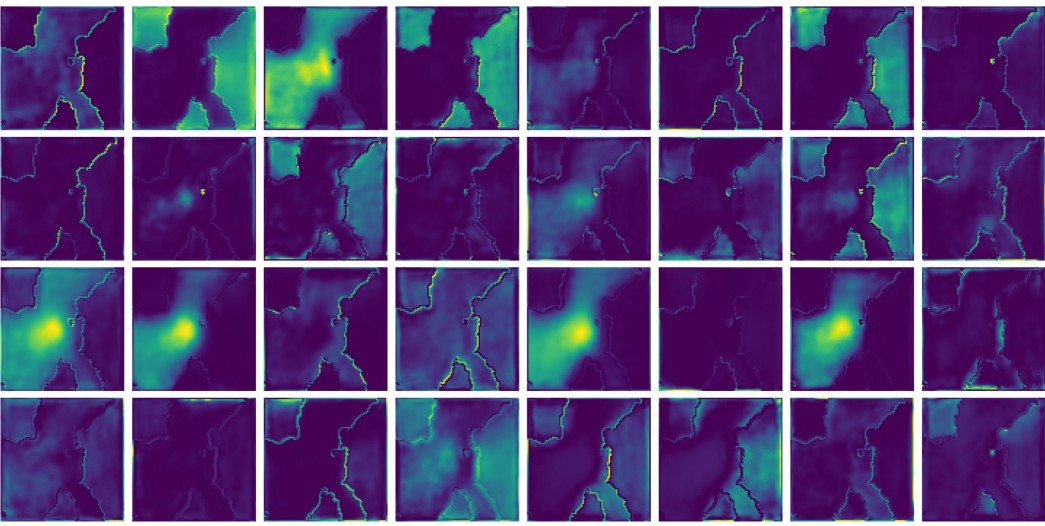

Figure 10: Pattern visualization of the 2D Darcy case. According to the discontinuities in the inputs, SKNO learns patterns at different scales.

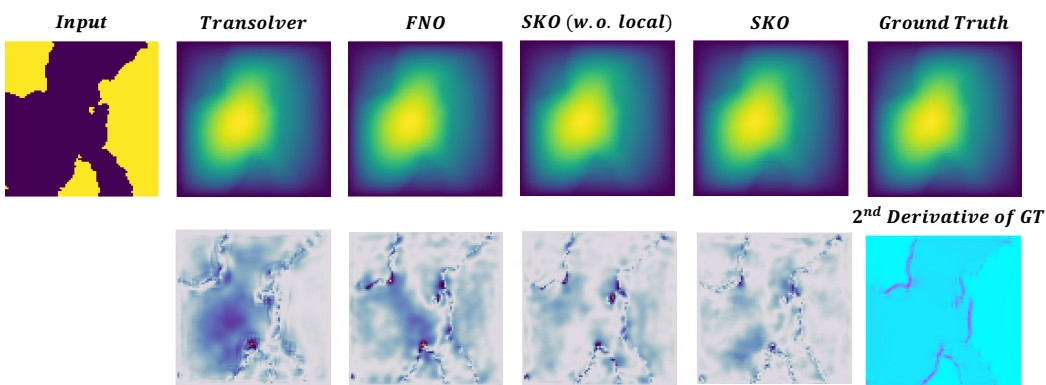

Figure 11: Prediction, error, and second derivative of ground truth visualization for 2D Darcy. The top row shows predictions from different models and ground truth, while the bottom row presents the corresponding error visualizations and the second derivative of the ground truth.

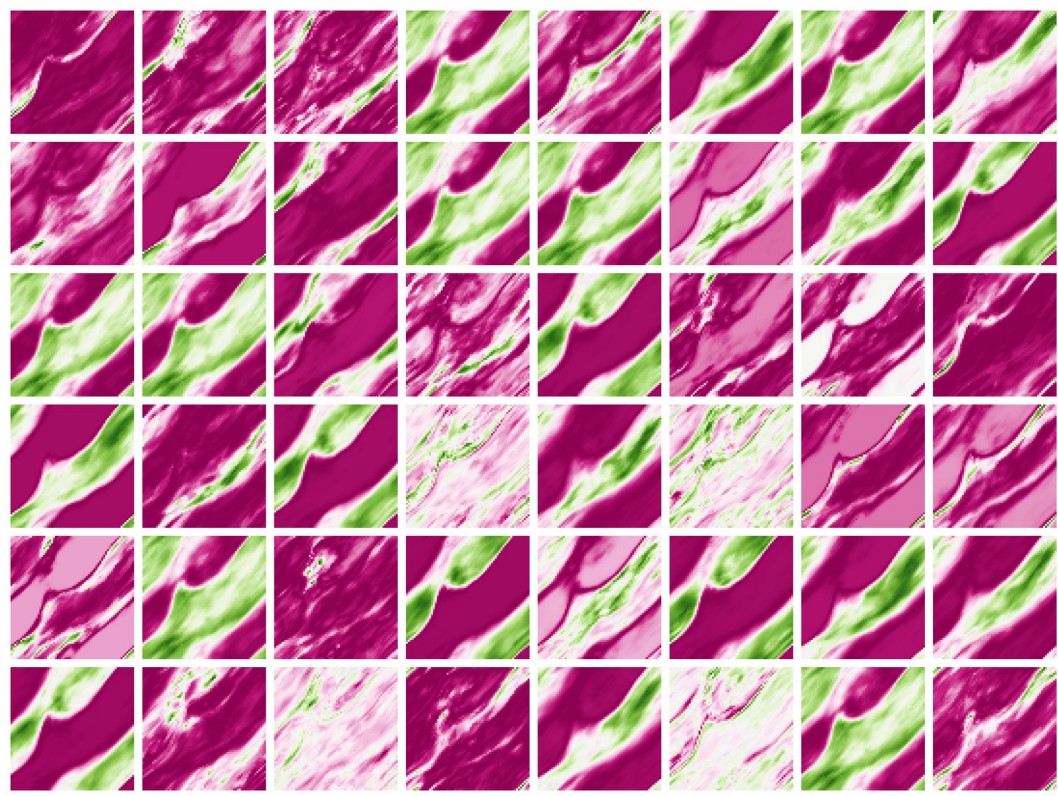

Figure 12: Pattern visualization of the 2D Navier-Stokes case.

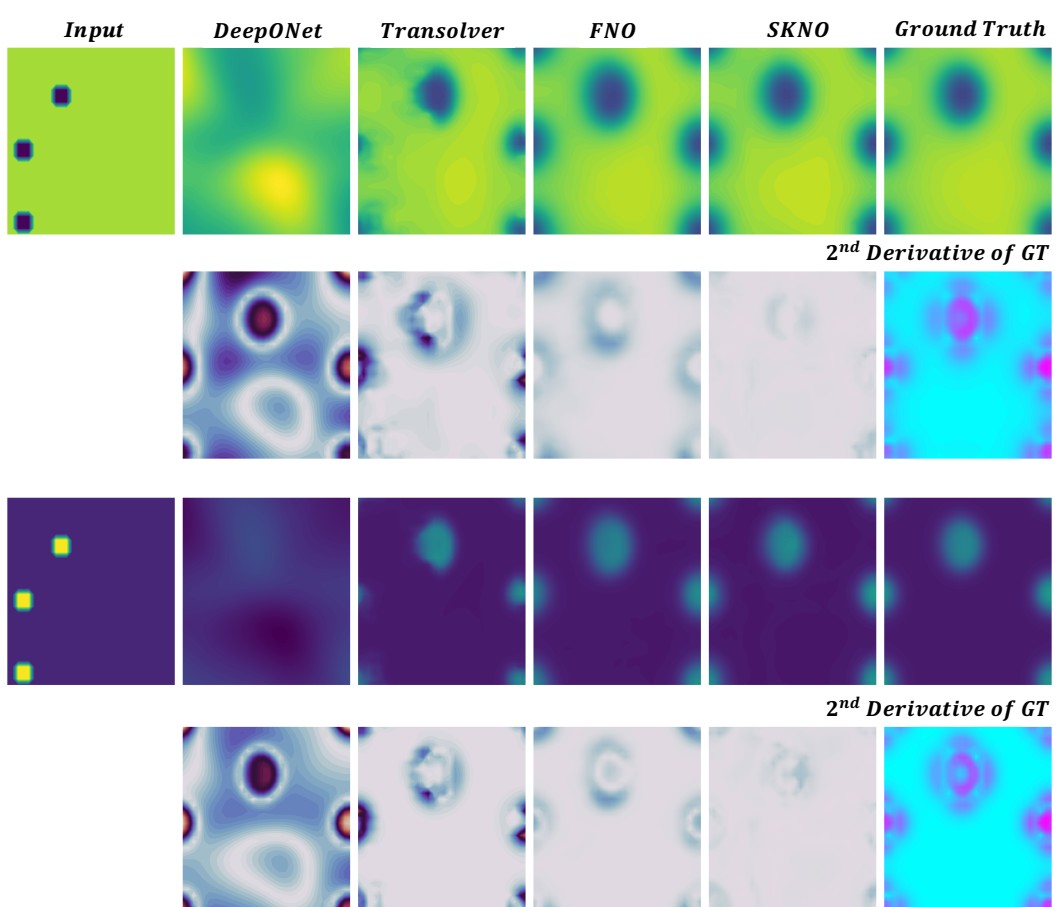

Figure 13: Visualization of the 2D Gray-Scott case. Top: $u$; Bottom: $v$. SKNO achieves the lowest error on both.

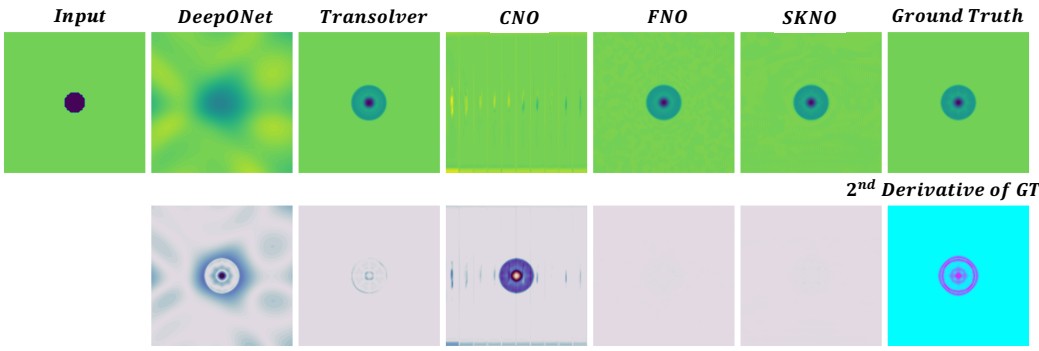

Figure 14: Visualization of the 2D Shallow-Water case with randomly mixed training data at different resolutions, evaluated at the highest resolution.

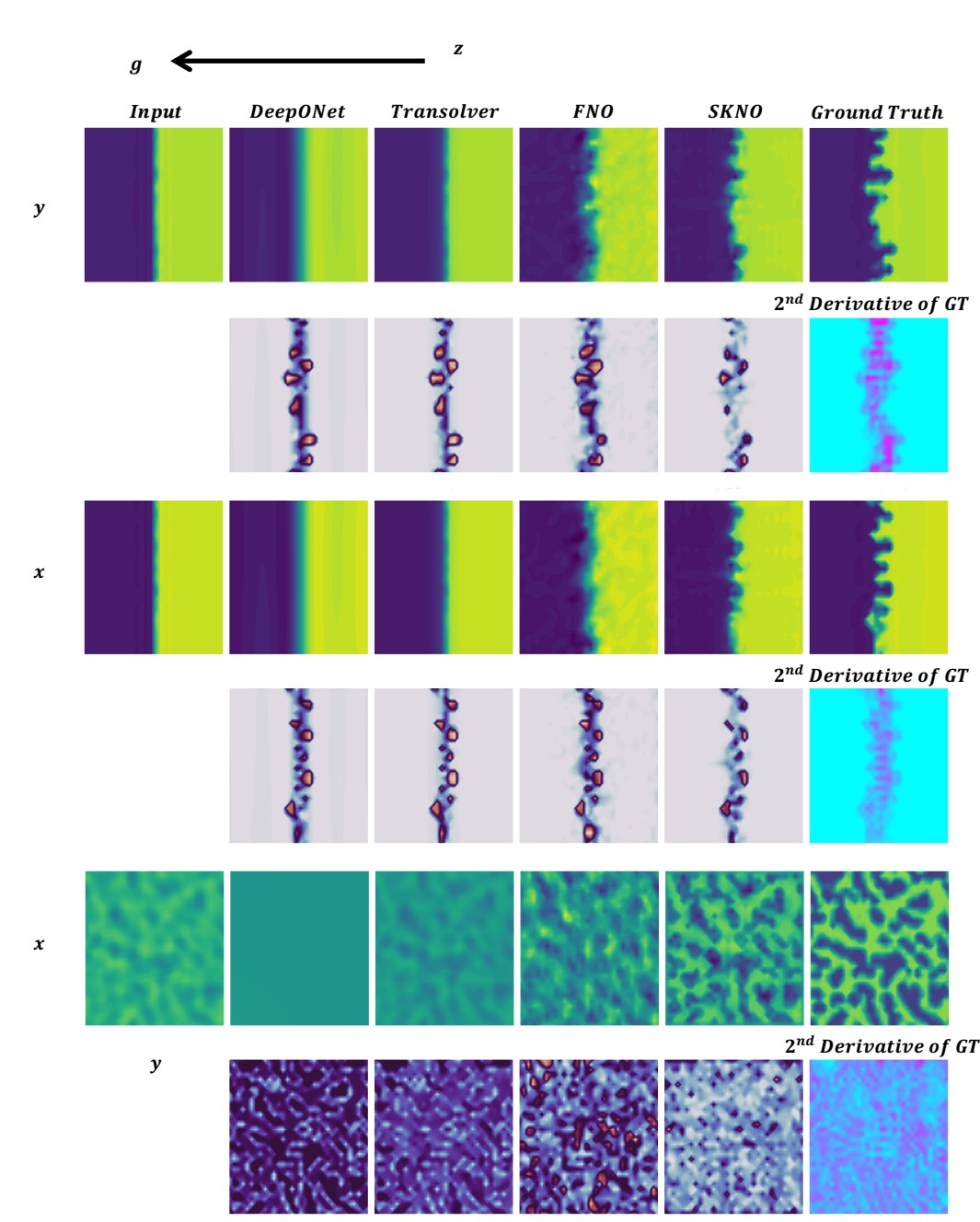

Figure 15: Visualization of the 3D Rayleigh–Taylor instability case. All slices are parallel to the coordinate planes, and their intersection points are chosen as the grid points nearest to the center of the box. We show a representative extreme case, namely predicting the output field after 20 steps from a perturbed initial condition. While other lightweight methods largely fail, SKNO can still capture the overall flow trends and the interfacial surfaces between the fluids with different densities. **g** represents the gravity direction.

## All 128 *Output Pattern fields in the Dictionary*
### (*Sorted by the absolute value of weights*)

### FNO

### SKNO

Figure 16: Visualization of paired output weights and pattern fields. All pattern fields are sorted in descending order of the absolute value of their associated output weights, and values marked in red lie below the contribution threshold of $10^{-3}$. Even with the same sufficiently large $N_p$, the relative $L_2$ error of SKNO on the testing data $6.546e-4$ is still larger than FNO $6.603e-4$.

