# OpenReview forum: "Redefining Neural Operators in $d+1$ Dimensions"
_ICLR.cc/2026/Conference — Submitted to ICLR 2026_

### Official Review · Reviewer_2PZZ · 2025-10-30

**Soundness:** 3
**Presentation:** 3
**Contribution:** 3
**Rating:** 6
**Confidence:** 3

**Summary:**

This paper proposes extending Neural Operators to an additional auxiliary dimension inspired by the Schrödingerisation method. By modeling operator evolution in this d+1-dimensional space, the authors design the Schrödingerised Kernel Neural Operator (SKNO), which combines spectral convolution and residual connections to capture both global and local dynamics. Although the theoretical foundation applies to linear PDEs, the model empirically performs well on nonlinear systems like Burgers and Navier–Stokes equations through stacked linear and nonlinear blocks. SKNO achieves state-of-the-art accuracy and resolution invariance across multiple PDE benchmarks.

**Strengths:**

- This paper is clearly structured, with a logical flow from theoretical motivation to model design, implementation, and experiments, making it easy to follow.
- The proposed approach is grounded in a solid theoretical framework based on the Schrödingerisation of linear PDEs, providing a principled explanation for the evolution mechanism within Neural Operators.
- The proposed SKNO achieves superior performance across ten diverse PDE benchmarks, demonstrating excellent accuracy, resolution invariance, and robustness compared to existing models.

**Weaknesses:**

- The introduction of Schrodingerisation is interesting, but the proposed model itself appears to simply add a one-dimensional input variable during lifting and then combine it with FNO and various existing ideas.
- The reason why the proposed method produces good results is somewhat unclear (See Questions).

**Questions:**

- Schrodingerisation is a technique for linear PDEs, and it seems that, inspired by this idea, a neural operator that can also be used for nonlinear PDEs was modeled. Is the reason why the proposed method performs well because it adds one dimension to the space to increase its expressiveness? Or is it because Schrodingerisation can be considered to incorporate the physical laws or mathematical structure from Hamiltonian system?

---

> ### Author Response · Authors · 2025-11-27
>
> We respond point by point.
>
> ---
>
> > Is the reason why the proposed method performs well because it adds one dimension to the space to increase its expressiveness? Or is it because Schrodingerisation can be considered to incorporate the physical laws or mathematical structure from Hamiltonian system?
>
> The latter is an inspiration of the former.
>
> The direct reason for the improved performance is increased model expressiveness. Schrödingerisation provides a principled way to introduce the auxiliary dimension by aligning classical linear PDEs with a unified Schrödinger-type formulation. And SKNO leverages this to model expressive components that previous neural operators ignored, leading to better approximation capacity and performance.
>
>
>
> > Simply add a one-dimensional input variable during lifting and then combine it with FNO and various existing ideas
>
> We acknowledge this, but we would like to emphasize our core contribution:
>
> Our work exposes and addresses a design bottleneck that has remained overlooked by the neural operator community since kernel integral neural operators were first proposed: the evolution of the PDE solution operator along the auxiliary dimension.
>
> We do not improve aggregation-based domain-transform operators (whether analytically well-understood or carefully engineered) in the original $d$-dimensional domain to gain performance. Instead, we show that existing methods can obtain substantial performance gains by adopting a $d+1$-dimensional operator design along this extra auxiliary dimension.
>
>
> **Additionally, we thank you for your positive assessment of both the clarity of our writing and the solid SKNO framework of our work.**

---

### Official Review · Reviewer_v7jn · 2025-10-31

**Soundness:** 2
**Presentation:** 2
**Contribution:** 2
**Rating:** 4
**Confidence:** 4

**Summary:**

This paper revisits the concept of Neural Operators (NOs) by formulating them explicitly in d + 1 dimensions, where the extra dimension represents temporal or parametric evolution. The authors argue that existing neural operators often neglect this intrinsic spatiotemporal coupling, which limits their ability to capture dynamic PDE behavior. The proposed framework offers a unified and physically grounded perspective that treats space and time consistently, leading to better representational power. Experiments on standard PDE benchmarks demonstrate notable improvements in accuracy over established baselines such as FNO and transformer-based models.

**Strengths:**

1) The paper provides a conceptually elegant and theoretically motivated redefinition of neural operators, naturally extending them to spatiotemporal systems.

2) It establishes a unified mathematical framework that connects spatial and temporal operator learning under a common perspective.

3) The proposed formulation achieves empirical improvements over strong baselines like FNO and Transformer-based architectures.

**Weaknesses:**

1) The paper is somewhat dense and difficult to follow in places — it took multiple readings to fully grasp the core formulation. Also it seems written in hurry as conclusion, future work, limitations aren't discussed.

2) Comparisons with newer operator paradigms, such as state-space models (e.g., Mamba Operator), are missing, which would provide a more comprehensive evaluation or consider citing them in related work.

3) The experimental scope is limited to low-dimensional PDEs, leaving scalability to 3D or real-world physics problems unexplored.

4) The computational cost and efficiency trade-offs of the proposed approach are not clearly discussed or quantified.

5) Some implementation details (e.g., architectural variations or training hyperparameters) could be presented more clearly for reproducibility.

**Questions:**

1) How does the proposed d + 1 formulation handle non-autonomous PDEs or systems with variable coefficients over time? Additionally, what happens if we extend this idea to even higher-dimensional formulations—would the dynamics become more linear in such limits (Koopman Operator), and have you experimented with this?

2) Have the authors evaluated the scalability of the approach on high-dimensional or 3D PDEs, and how does its computational cost grow with problem size?

3) Could the proposed framework be combined with latent-space modeling or physics-informed priors to enhance interpretability and regularization?

4) How does the model perform under zero-shot generalization to unseen temporal regimes or boundary configurations?

5) In Figure 3, it’s unclear why multiple global propagators and only one local propagator were used. Have different configurations been tested, and how do they affect performance?

**Missing References:**

1) Complex Neural Operator: https://arxiv.org/abs/2406.02597

---

> ### Author Response · Authors · 2025-11-27
>
> We respond point by point.
>
> ---
>
> > dense and difficult to follow
>
> Because we need to distinguish tensor multiplications, two types of dimensions, and their corresponding Fourier domains, the notations seem to be dense. In the revised version, we have updated the notation table (*Table 6*), and we will further simplify the notations in our future version.
>
> > conclusion, future work, limitations aren't discussed.
>
> In the submitted version, the conclusion has already appeared in *lines 1147–1159*.
>
> In the revised version, we have added an explicit discussion of future work and limitations, and we have moved this section to the end of the main text.
>
> > Missing References
>
> We have updated *Appendix D* to include discussion and citations for these two works. While CoNO and LaMO introduce effective inductive biases in the original $d$-dimensional domain to improve performance, SKNO demonstrates that breaking the bottleneck along the auxiliary embedding dimension brings substantial gains.
>
> > leaving scalability to 3D or real-world physics problems unexplored (Have the authors evaluated the scalability of the approach on high-dimensional or 3D PDEs?)
>
> In the submitted version, we have already evaluated SKNO on these problems: 3D compressible Navier–Stokes equations, 3D Rayleigh–Taylor instability, and global wind prediction on ERA5. The corresponding results and discussions can be found originally in *Section 4.2 - Results*, *Appendix C - Benchmarks and Results - Items 8–10*, *Tables 3, 10, 11, 12*. (3, 11, 12, 13 in the revised version) and *Figure 12* (14 in the revised version).
>
> > The computational cost and efficiency trade-offs of the proposed approach are not clearly discussed or quantified.
>
> We have provided a detailed analysis of computational complexity and its scaling with problem size in *Appendix E – Complexity Analysis* of our submitted version. This discussion is further clarified and maintained in the *Appendix E* of our revised version.
>
> > Some implementation details (e.g., architectural variations or training hyperparameters) could be presented more clearly for reproducibility.
>
> In the submitted version, we have reported the hyperparameter settings in *Table 7* and described the configurations of some other models in *Appendix C - Baselines*. Since our claimed principle is *"to keep each baseline at the best configuration recommended in its original paper or code, rather than forcing all methods to share a single fixed setting"*, almost all other hyperparameters are kept at their default values reported by their papers or published codes. For SKNO, we share the same hyperparameter settings as FNO where applicable.
>
> We have added a more detailed list of model configurations in the supplementary materials, including possible architectural variations or training hyperparameters. If there are still concerns about reproducibility, we kindly refer the reader to the code provided in the supplementary material.
>
> > How does the proposed d + 1 formulation handle non-autonomous PDEs or systems with variable coefficients over time?
>
> In the submitted version, we have already tested SKNO's SOTA performance on two challenging non-autonomous PDEs: 3D compressible Navier–Stokes equations and 3D Rayleigh–Taylor instability. These setups involve time-dependent dynamics and variable coefficients, and the corresponding descriptions and results can be found in *Section 4.2*, *Appendix C - Benchmarks and Results - Items 8 and 9*, and *Table 3, 11*.
>
> > What happens if we extend this idea to even higher-dimensional formulations—would the dynamics become more linear in such limits (Koopman Operator), and have you experimented with this?
>
> This raises the issue of the curse of dimensionality. To make such extensions practical, one likely needs additional structure, such as extra regularization mechanisms, to maintain the sparsity of search spaces and converge to correct parameters in high-dimensional spaces. We have not yet systematically experimented with higher-dimensional formulations whose absence does not affect the contributions of the current version. And we plan to work on this as one of our future research directions.
>
> > Could the proposed framework be combined with latent-space modeling or physics-informed priors?
>
> The bottleneck we have highlighted existed in almost all previous kernel integral neural operator designs. Taking LaMO as an example: although it employs multi-head attention, it does not impose a specific "operator" inductive bias along the embedding dimension corresponding to $p$. Under the $d+1$-dimensional viewpoint, a possible extension is to inject the latent-space modeling of LaMO also along the auxiliary $p$-dimension, based on how such biases are used in the original $d$-dimensional domain.
>
> > How does the model perform under zero-shot generalization to unseen temporal regimes?
>
> We have shown the SOTA performance of SKNO on 2D incompressible Navier-Stokes in *Fig.8* of the newly revised version.

---

> ### Author Response · Authors · 2025-11-27
>
> > How does its (SKNO's) computational cost grow with problem size?
>
> Beyond the high memory cost of storing high-resolution 3D data itself, the model-side memory of an SKNO under a fixed parameter configuration does not change.
>
> > Why were multiple global propagators and only one local propagator used? Different configurations?
>
> For the global propagators, decreasing \ increasing the number of global propagators will increase \ decrease the model’s relative $L_2$ error on the test data, which has similar performance to FNO.
>
> One local propagator is introduced to aggregate information in local regions and fine-tune predictions where the residual is large, compensating for the smoothing and low-frequency bias induced by spectral convolutions in the global propagators. Here we compare configurations with zero, one, and two local propagators, and show that adding local aggregation layers helps the $d+1$-dimensional signal integrate residual information more effectively with a lower level of entanglement (*Entanglement Entropy: A measure of the degree of entanglement between $d$-dim and auxiliary-dim subsystems of a composite quantum system*).
>
> | Darcy                            | Without Local Prop | With 1 Local Prop | With 2 Local Props |
> |-------------------------------------|--------------|-------------|------------------------|
> | Relative $L_2$ Error                         | 5.717        | 5.555       | 5.289                  |
> | Entanglement Entropy (Before Recovering)   | 2.286      | 1.849       | 1.277                  |

---

### Official Review · Reviewer_QUDn · 2025-10-31

**Soundness:** 2
**Presentation:** 2
**Contribution:** 2
**Rating:** 4
**Confidence:** 5

**Summary:**

This paper introduces a novel framework for Neural Operators (NOs) by reformulating them in a d+1 dimensional space. The core motivation stems from the "Schrödingerisation" method in quantum physics, which transforms a d-dimensional PDE into a d+1 dimensional equivalent. The authors propose that this higher-dimensional perspective can elucidate and improve the design of NOs. Based on this framework, they develop the Schrödingerised Kernel Neural Operator (SKNO), a model that explicitly operates in this d+1 space. SKNO consists of three stages: a lifting operator to map the input to the higher-dimensional space, an evolution block that propagates information in both the original d dimensions and the new auxiliary dimension, and a recovering operator to project the result back to the original d-dimensional space. The authors conduct extensive experiments across ten PDE benchmarks, claiming that SKNO achieves state-of-the-art performance, superior resolution invariance, and better zero-shot super-resolution capabilities compared to existing NOs like FNO and Transolver.

**Strengths:**

The paper introduces a creative, physics-inspired viewpoint for designing neural operators, which could stimulate new research directions.

**Weaknesses:**

**Weaknesses***

1.  **Weak Theoretical Grounding:** The foundational analogy to Schrödingerisation feels more inspirational than formal. The paper lacks a rigorous derivation showing why this `d+1` formulation is a necessary or uniquely optimal way to construct neural operators, weakening the claim of having "redefined" them.
2.  **Unclear Computational Cost:** The paper does not adequately address the increased computational complexity of its approach. The use of a `(d+1)`-dimensional FFT is significantly more expensive than the `d`-dimensional FFT in FNO. A transparent comparison of parameter counts and FLOPs is missing, which is crucial for evaluating whether the reported accuracy gains justify the additional computational burden.
3.  **Incremental Architectural Novelty:** When stripped of its quantum physics motivation, the SKNO architecture can be seen as a multi-channel FNO variant where channels (the auxiliary `p` dimension) interact through an additional learned operator. The building blocks are well-established, making the contribution more of an effective architectural design than a fundamental paradigm shift.
4.  **Limited Justification for Design Choices:** The paper implements the evolution along the auxiliary dimension using a spectral operator. The rationale for this specific choice over other possible mechanisms (e.g., attention, simple MLPs) is not thoroughly explored, leaving it unclear if this is a critical component of the framework's success.

5. **Some baselines' performance[1] is great, but authors does't compared**

[1]. Turb-L1: Achieving Long-term Turbulence Tracing By Tackling Spectral Bias

**Questions:**

1.  Could you provide a more formal justification for the `d+1` framework? Beyond the analogy, is there a theoretical reason (e.g., from an approximation theory perspective) why adding an auxiliary dimension and evolving along it should lead to better performance for learning PDE solution operators?
2.  For a representative benchmark like the 2D Navier-Stokes or 3D Rayleigh-Taylor instability, could you please provide a direct comparison of the total number of trainable parameters and the estimated FLOPs per forward pass for SKNO, FNO, and Transolver with the configurations used in your experiments?
3.  How does the performance of SKNO change as you vary the resolution of the auxiliary dimension, `N_p`? The paper seems to fix `N_p` based on the baseline (e.g., Table 7). Is there a trade-off between accuracy and computational cost associated with this hyperparameter?
4.  The evolution along the `p`-dimension is a key component. Have you experimented with replacing the spectral operator (`F^-1 * A * F + b`) with a simpler or different type of operator? For instance, how would a simple MLP applied across the `p`-dimension "channels" perform? This would help isolate the benefits of the `d+1` structure from the specific choice of operator.

---

> ### Author Response · Authors · 2025-11-27
>
> We respond point by point.
>
> ---
>
> > More inspirational than formal: why this $d+1$ formulation is a necessary or uniquely optimal way to construct neural operators. / a more formal justification for the $d+1$ framework
>
> In *Appendix E* of our revised version, we add experiments and extra formal discussions on the sparse representations of SKNO to further demonstrate the advantages of this design. What needs to be clarified is, we **do not** claim that it is a uniquely optimal way to construct neural operators. Honestly, no model in the current stage can claim to be the uniquely optimal neural operator architecture.
>
> Our contribution is to uncover a design dimension that has been ignored in previous kernel integral neural operator designs: instead of improving aggregation-based transforms on the $d$-dimensional domain, we discuss another neglected bottleneck of the current operator framework inspired by the Schrödingerisation. After redefining a $d+1$-dimensional framework, an implementation called SKNO shows strong advantages.
>
> In *Section 5* of our revised version, we also explicitly acknowledge the limitation of quantitatively characterizing and comparing the general approximation errors of different architectures, which is still a challenging problem for other designs of neural operators and related theoretical works.
>
> > The use of a $d+1$-dimensional FFT is significantly more expensive than the $d$-dimensional FFT in FNO.
>
> Not "significantly more expensive". The kernel integration of SKNO has linear complexity with the 1-2x constant as FNO, while the linear-attention-based Transolver at least larger than 4x. We have discussed the complexity difference between SKNO and FNO in detail in *Appendix E – Complexity Analysis* of both the submitted and revised versions, where we make explicit how the $d+1$-dimensional FFT affects the overall scaling and where it sits relative to FNO and Transolver.
>
> > Comparison of parameter counts and FLOPs is missing. (For a representative benchmark like the 2D Navier-Stokes or 3D Rayleigh-Taylor instability, could you please provide a direct comparison of the total number of trainable parameters and the estimated FLOPs per forward pass for SKNO, FNO, and Transolver with the configurations used in your experiments?)
>
> Under the configurations used in our submitted paper and code, we report the FLOPs of the kernel integration for these experiments below:
>
> | Model | FNO | Transolver | SKNO |
> |---|---|---|---|
> | **2D NS** ||||
> | FLOPs | 1.16e8 | 1.19e9 | 2.12e8 |
> | Rate | 1.000 | 10.292 | 1.831 |
> | **3D R–T Instability** ||||
> | FLOPs | 8.98e8 | 9.00e9 | 1.59e9 |
> | Rate | 1.000 | 10.029 | 1.774 |
>
> which is consistent with the formulas in *Table 8*.
>
> > The contribution more of an effective architectural design than a fundamental paradigm shift.
>
> We think that the claim of a “fundamental paradigm shift” should be reserved for future validation by the community, and we have not presented our contribution in those terms. In the submitted version, we described our work as *"providing a new perspective to understand the evolving mechanism in neural operators"*.

---

> ### Author Response · Authors · 2025-11-27
>
> > The rationale for this specific choice over other possible mechanisms (e.g., attention, simple MLPs) is not thoroughly explored. / Have you experimented with replacing the spectral operator with a simpler or different type of operator? / How would a simple MLP applied across the $p$-dimension "channels" perform?
>
> As *Problem 2* in the introduction of the submitted version, an MLP along the auxiliary dimension is essentially what the previous methods have done (a matrix plus nonlinearity on channels). Our choice of a spectral operator is motivated by the Schrödingerisation view and has been evaluated against MLP-style baselines. Attention-based or other learnable aggregation operators along the auxiliary dimension are allowed to replace the spectral choice in SKNO and can be incorporated into the same $d+1$ framework, where we have evaluated our SKNO design following the introduced Schrödingerisation method.
>
> If you are interested in the performance of "an MLP-based method" and ours, you can see the comparisons between SKNO and FNO in *Table 3, 8-14* and *Figure 5-7, 10, 12-15*.
>
> In addition, we show the results on 2D Darcy with other possible implementations (under both $d$-dimensional and $d+1$-dimensional frameworks) below:
>
> |Model|FNO|2Stack-FNO|SKNO-Attention|SKNO (Ours)|
> |---|---|---|---|---|
> |Rel. $L_2$ Err.|0.0062| 0.0061  | 0.0058 |**0.0055**|
> |Training Time| **10.21 min** |13.35 min| 76.90 min |17.59 min|
>
> where (1) the original SKNO spectral implementation following the introduced Schrödingerisation method achieves the best performance, and (2) the right two $d+1$ dimensional NO implementations perform better than the left two $d$ dimensional implementations. The two variants:
>
> - 2Stack-FNO: simply stacking two Fourier Layers in the $d$-dimensional setting, which ignores the operator design along $p$.
>
> - SKNO-Attention: replace the spectral operator in SKNO with attention-based aggregation without explicitly aligning the $d+1$ dimensional Schrödingerised form.
>
>
> > Some baselines' performance[1] is great, but authors does't compared
>
> In the referenced paper, four operator-learning models are compared. In our submitted version, we have already included two of these as baselines. Here, we additionally add one of the well-implemented baselines in this paper and summarize the comparisons on 2D Darcy below:
>
> |Model|DeepONet|GNOT|OFormer|LSM|CNO|FNO|Transolver|SKNO (Ours)|
> |---|---|---|---|---|---|---|---|---|
> |Rel. $L_2$ Err.|0.0609|0.0102|0.0121|0.0065|0.0060|0.0062|0.0058|**0.0055**|
> |Training Time|**5.20 min**|22.58 min|21.91 min|20.37 min|63.34 min|10.21 min|115.48 min|17.59 min|
>
> We have also added corresponding citations for all four operators that this paper has mentioned in the revised version.
>
> > How does the performance of SKNO change as you vary the resolution of the auxiliary dimension, $N_p$?
>
> In our experiments, SKNO uses the same small $N_p$ as FNO to ensure a fair comparison.
>
> If we increase $N_p$, SKNO gains a larger dictionary capacity to capture finer-scale physical pattern fields. As illustrated in the bottom–middle/right panels of *Fig. 6* and in *Fig. 15* of our revised version, increasing $N_p$ enlarges the dictionary with pattern fields from large scales to small scales.
>
> > The paper seems to fix $N_p$ based on the baseline (e.g., Table 7). Is there a trade-off between accuracy and computational cost associated with this hyperparameter?
>
> For fair comparison, SKNO adopts the same $N_p$ setting as FNO to make sure it cannot gain extra advantages from a larger $N_p$. Transolver, in turn, uses its default best $N_p$ choices reported in its original work and codes.
>
> As with any previous neural operators, there is always a trade-off between accuracy and computational cost of SKNO. The recommended settings we adopt for each baseline (including SKNO) typically lean towards achieving better accuracy under reasonable cost, consistent with their original papers.

---

### Official Review · Reviewer_YsY9 · 2025-11-01

**Soundness:** 2
**Presentation:** 2
**Contribution:** 2
**Rating:** 4
**Confidence:** 4

**Summary:**

This paper tackles the problem of Neural Operators from the Schrodingerisation method used in quantum simulations of PDEs. Hoping to address the use of discretization matrix in the current methods, it proposes a new operator, namely Schrodingerised Kernel Neural Operator (SKNO), which is claimed to be operating on a (d+1) dimensional space. This is claimed to be better aligned with the underlying evolution mechanism. Experiments are conducted on a range of datasets to show the efficacy of the method.

**Strengths:**

1. Showing the connection between  Schrodingerisation and NO is interesting.

2. It is shown that SKNO is resolution invariant.

3. Good experimental design.

**Weaknesses:**

1. There is no strong intuition for why Schrodingerisation should be the best choice for the NO (beyond empirical success).

2. Theorems 1 and 2 effectively restate the FNO results in (d+1) dimension rather than showing why this formulation is fundamentally different/better (in terms of convergence, scale, computations etc) than the exisisting methods.

3. The architectural design seems incremental compared to FNO (except for the auxiliary dimension).

4. Computational complexity claims are not accurate (the additional log N_p factor seems to be downplayed).

5. Looks like the method is over-sensitive to the hyper-parameter choices.

6. Chosen baselines seem limited (see this for more datasets and methods https://arxiv.org/pdf/2310.01650). Further, transolver and SKNO use different levels of N_p.

7. The paper uses very dense notations (both in the text and figures), which makes it a little hard to read.

**Questions:**

1. Can you think of a theorem showing that d+1 dimensional operators have greater expressiveness than d-dim ones for a given parameter budget?

2. Is there a physical interpretation for the auxiliary dimension p?

3. What happens if N_p is increased to match that of Transolver?

4. Despite SKNO being designed for linear PDEs, how can it do well on highly non linear problems?

5. Can an iso-paramter/iso-flop comparison be shown?

6. Is there a principled way of making choices in Tab 5, it seems arbitrary now.

7. Are there scenarios where SKNO performs worse than baselines? When does d+1 framework not help?

8. For 3D problems with finer resolution, how does memory scale with N_p?

---

> ### Author Response · Authors · 2025-11-27
>
> We respond point by point.
>
> ---
>
> > Intuition for why Schrodingerisation should be better than the existing methods / why $d+1$ dimensional operators have greater expressiveness than $d$-dim ones (under a given parameter budget)
>
> We have added experiments and corresponding proof in *Appendix E - Sparse Representation* of the revised version to show the advantage of SKNO in sparse representations: under the same $N_p$, $d+1$ dimensional SKNO admits a more energy-concentrated sparse representation than FNO.
>
> We also explicitly acknowledge in *Section 5* of the revised version that it is still challenging to quantitatively compare the general approximation errors of different architectures, which remains a limitation not only for our work but also for others (both theoretical works and specific neural operators).
>
> > Is there a physical interpretation for the auxiliary dimension $p$?
>
> In SKNO, $p$'s corresponding index $\eta$ in $p$'s Fourier domain can be interpreted as the index of physical scales. From the Schrödingerisation viewpoint shown in *Example 1*, $\eta$ indices different “effective Planck constants”.
>
> > The architectural design seems incremental compared to FNO (except for the auxiliary dimension).
>
> We acknowledge this, but we would like to emphasize our core contribution:
>
> Our work exposes and addresses a design bottleneck that has remained overlooked by previous neural operators since kernel integral neural operators were first proposed: the evolution of the PDE solution operator along the auxiliary dimension.
>
> Rather than following existing formulations and improving aggregation-based domain-transform operators (whether analytically well-understood or carefully engineered) in the original $d$-dimensional domain to gain performance, we show that existing methods like FNO can obtain substantial performance gains by adopting a $d+1$-dimensional operator design along this extra auxiliary dimension.
>
> > Computational complexity claims are not accurate (the additional $\log N_p$ factor seems to be downplayed).
>
> In *Appendix E* of the revised version, we explicitly compute the additional scale factor $\log N_p$, where $\log⁡ N_p=5$ is treated as an additive constant to FNO’s $\log ⁡N_x=27$ in that high-resolution 3D case.
>
> Hence, the SKNO kernel implementation keeps a complexity within a 1–2x constant factor of FNO. For the same depth, this remains at least 2–4x lower than the Transolver complexity (scale factor $>128$) in the same setting.
>
> > Looks like the method is over-sensitive to the hyper-parameter choices.
>
> The hyperparameter sensitivity of SKNO is comparable to that of FNO, which is known for its broad applicability and relatively robust hyperparameter tolerance (i.e., within a reasonable range of settings, the model still converges well with errors in the same order of magnitude).
>
> As in FNO, SKNO is mainly sensitive to two hyperparameters: the number of truncated modes and $N_p$. Other model hyperparameters can be kept at their recommended default settings across test scenarios without tuning.
>
> > More datasets and methods (arXiv paper)
>
> - Method: Our baseline comparisons in the submitted version have covered the choosen neural operators discussed in the paper you mentioned, and four of the PDE scenarios in that paper are also evaluated in our work. For 2D Darcy,
>
> |Model|DeepONet|GNOT|OFormer|LSM|CNO|FNO|Transolver|SKNO (Ours)|
> |---|---|---|---|---|---|---|---|---|
> |Rel. $L_2$ Err.|0.0609|0.0102|0.0121|0.0065|0.0060|0.0062|0.0058|**0.0055**|
> |Training Time|**5.20 min**|22.58 min|21.91 min|20.37 min|63.34 min|10.21 min|115.48 min|17.59 min|
>
> - Dataset: We have also evaluated SKNO's SOTA performance on the 2D STRESS and 2D STRAIN benchmarks in your provided paper, which has been cited and added to *Table 14* in the revised version.
>
> > Transolver and SKNO use different levels of N_p? / What happens if N_p is increased to match that of Transolver?
>
> In our experiments, SKNO uses the same small $N_p$ as FNO to ensure a fair comparison.
>
> If we increase $N_p$, SKNO gains a larger dictionary capacity to capture finer-scale physical pattern fields. As illustrated in the bottom-middle/right panels of *Fig. 6* and in *Fig. 15* in our revised version, increasing $N_p$ effectively enlarges the dictionary from large-scale to small-scale patterns.
>
> > Dense notations
>
> Because we need to distinguish tensor multiplications, two types of dimensions, and their respective Fourier domains, the notation is inevitably dense. We have updated *Table 6* to clarify some key notation usages and will try to simplify them further in the future version.

---

> ### Author Response · Authors · 2025-11-27
>
> > Can an iso-paramter/iso-flop comparison be shown?
>
> Under the configurations used in our submitted paper and codes, we report the FLOPs of the kernel integration for the challenging experiments below.
>
> | Model | FNO | Transolver | SKNO |
> |---|---|---|---|
> | **2D NS** ||||
> | FLOPs | 1.16e8 | 1.19e9 | 2.12e8 |
> | Rate | 1.000 | 10.292 | 1.831 |
> | **3D R–T Instability** ||||
> | FLOPs | 8.98e8 | 9.00e9 | 1.59e9 |
> | Rate | 1.000 | 10.029 | 1.774 |
>
> which is consistent with the complexity analysis shown in *Table 8* of both the submitted and the revised version.
>
> > Is there a principled way of making choices in Tab 5, it seems arbitrary now.
>
> Not arbitrary. As clarified in the submitted version (*lines 933–950*), our principle is *"to keep each baseline at the best configuration recommended in its original paper or code, rather than forcing all methods to share a single fixed setting."*
>
> For example, SKNO uses the same best $N_p$ setting as FNO, while Transolver uses the default best $N_p$ setting reported in its original work. Detailed descriptions have been provided in *Sec. 4.1 - Baselines and Configurations* and *Appendix C - Baselines* in our submitted version.
>
> > For 3D problems with finer resolution, how does memory scale with $N_p$?
>
> Beyond the high memory cost of storing high-resolution 3D data itself, the model-side memory of an SKNO under a fixed parameter configuration does not change.
>
> > Are there scenarios where SKNO performs worse than baselines? When does $d+1$ framework not help?
>
> Under extremely small $N_p$ or very limited data.

---

### Author Response · Authors · 2025-12-03
**Review and Rebuttal Summary**

According to the original reviews, we and the reviewers have reached a consensus at least on:

* Interesting connections between Schrödingerisation and Neural Operators; broad and good experiments (*Reviewer YsY9*)
* A creative, physics-inspired viewpoint and a novel framework for Neural Operators, stimulating new research directions; extensive experiments (*Reviewer QUDn*)
* A conceptually elegant and theoretically motivated redefinition of neural operators; notable improvements in accuracy (*Reviewer v7jn*)
* State-of-the-art accuracy, resolution invariance, and robustness; clearly structured and easy to follow; a solid theoretical framework based on the Schrödingerisation (*Reviewer 2PZZ*)

---

During the rebuttal stage, we have responded to all questions and comments on weaknesses from the reviewers.

A brief summary is as follows:

- **Why SKNO performs better** (e.g. *Reviewer QUDn, Q1*)

    - We have added additional experiments and corresponding formal proofs to show readers the sparser representation of SKNO.

    - We have also explicitly acknowledged that providing a general, quantitative approximation error relationship for different neural operators still remains a challenge for our work, other neural operator designs, and related theoretical works. We will further investigate this in our future theoretical work.

* **Clarifications on computational complexity and hyper-parameter choices** (e.g. *Reviewer YsY9, W4 & Q6*)

  * **Computational complexity**: We have shown exact calculations in our complexity analysis, which are concrete and cause no confusion. Additionally, the newly reported FLOPs are consistent with our original analysis.
  * **Hyper-parameter choices**: We have highlighted the experimental protocol already stated in the submitted version: *under the constraint of fairness (e.g., no prior regularization in the loss function, key parameters of FNO and SKNO kept consistent, etc.), we respect the best settings of each baseline reported from their papers and codes, rather than forcing all configurations (e.g., optimizer, network width/depth, positional encoding, etc.) into a single fixed template.*

* **Other questions** (e.g. *Reviewer YsY9, W6* and *Reviewer v7jn, Q5*)

  * **New citations, benchmarks, and model comparisons given by reviewers**: all requested references, benchmarks, and comparisons have been addressed in the point-by-point responses and incorporated into the revised version.
  * **Additional experiments and model performance that specific reviewers were interested in**: All of them have been conducted and reported, and the corresponding questions have been answered based on these results.

Most of the remaining concerns and weak points from reviewers overlooked contents already presented in the submitted version (e.g. *Reviewer v7jn, Q2*), which we have provided direct, point-by-point clarifications. We believe that for readers who carefully read the paper, these issues can be fully resolved by the submitted version.

---

***Many thanks to all members of the ICLR community who have contributed to reviewing our paper; we hope to work together to foster a healthy, virtuous ICLR.***

---

### Meta-Review · Area_Chair_GKfm · 2026-01-07

**Summary:**

Based on the review discussion, here's a summary of the key concerns that emerged:
1. Multiple reviewers (YsY9, QUDn, v7jn) noted that while the Schrödingerisation connection is intellectually interesting, the paper doesn't provide rigorous theoretical grounding for why the d+1 formulation should fundamentally outperform d-dimensional approaches. Reviewer QUDn characterized it as "more inspirational than formal," and several asked for formal approximation theory arguments explaining the expressiveness gains.
2. Reviewers YsY9 and QUDn raised questions about whether the accuracy improvements justify the additional computational burden of d+1 dimensional FFTs. While the authors provided FLOPs comparisons during rebuttal showing SKNO remains within 1-2x of FNO's complexity, this wasn't adequately addressed in the original submission. The "additional log N_p factor" concern suggests reviewers felt this cost was understated.
3. Reviewer QUDn's assessment that SKNO is essentially "a multi-channel FNO variant where channels interact through an additional learned operator" captures a recurring theme - that stripped of the physics motivation, the architectural novelty appears modest. The core building blocks remain established components from prior work.
4. Reviewers v7jn and YsY9 found the notation dense and the paper difficult to follow. Reviewer v7jn noted the paper "seems written in hurry" with missing discussions of conclusions, limitations, and future work in the original submission.
5. Reviewer YsY9 raised concerns about the method appearing "over-sensitive to hyperparameter choices" and questioned whether comparison protocols were fair, noting different N_p values used across baselines.
6. Several reviewers identified gaps in experimental comparisons with recent methods and datasets that should have been included.

Despite these concerns, all reviewers acknowledged the extensive experimental validation, consistent performance improvements across benchmarks, and the conceptual elegance of connecting Schrödingerisation to neural operator design. The authors' rebuttal addressed many technical questions, though the fundamental theoretical justification gap remained.

**Reviewer Concerns:**

Concerns Adequately Addressed

Reviewers YsY9 and QUDn raised computational complexity concerns that were effectively resolved through concrete FLOPs calculations showing 1.7-1.8x overhead versus FNO and 5-6x efficiency versus Transolver. The detailed Appendix E analysis demonstrates the cost remains manageable despite the additional dimension.

The missing baselines and benchmarks identified by QUDn and v7jn were comprehensively addressed. Authors added comparisons with DeepONet, GNOT, OFormer, LSM, and CNO, included new 2D STRESS/STRAIN benchmarks, and incorporated relevant citations for recent work like Complex Neural Operator.

Reviewer YsY9's concerns about hyperparameter sensitivity were clarified through explanation of the experimental protocol. The authors respect each baseline's optimal settings from original papers rather than forcing uniform configurations, which is methodologically sound.

Reviewer v7jn's scalability concerns appear to stem from overlooking existing content. The authors effectively pointed to 3D experiments on compressible Navier-Stokes and Rayleigh-Taylor instability already present in the submission.

Design choice questions from QUDn and v7jn received support through ablation studies comparing spectral versus attention versus simple stacking approaches, plus local propagator analysis using entanglement entropy metrics.

Concerns Insufficiently Addressed

The fundamental theoretical justification gap identified by multiple reviewers remains unresolved. While authors added sparse representation experiments, they concede that quantitatively comparing general approximation errors across architectures remains challenging. The response amounts to empirical validation rather than rigorous approximation theory proving why the d+1 formulation is necessary or optimal. The connection between Schrödingerisation and improved performance stays more inspirational than formally grounded.

Reviewer YsY9's question about physical interpretation of the auxiliary dimension received only vague answers. Describing η as representing "physical scales" or "effective Planck constants" lacks clear intuition for what this dimension means in practical PDE solving contexts beyond mathematical convenience.

A critical gap concerns why this method works on nonlinear PDEs. Both YsY9 and 2PZZ noted the disconnect between motivation from linear PDE Schrödingerisation and strong empirical performance on highly nonlinear problems like Navier-Stokes. The claim that stacking linear and nonlinear blocks handles nonlinearity lacks mechanistic justification.

Reviewer QUDn's assessment that this represents an incremental contribution rather than a paradigm shift was not convincingly rebutted. The authors acknowledge not claiming a fundamental shift but don't adequately address whether SKNO is essentially multi-channel FNO with learned channel interactions. The framing as exposing a "design bottleneck" may overstate what amounts to an effective engineering improvement.

Presentation concerns persist despite revisions. Authors acknowledge notation remains "inevitably dense" even after Table 6 updates, leaving legitimate accessibility barriers for the broader community. Reviewer v7jn's observation about rushed writing and missing discussions was only partially resolved.

**Reviewer Scores:**

Reviewer YsY9 (Initial: 4)
Would likely maintain score at 4 or increase to 6. The reviewer's major technical concerns about computational complexity and hyperparameter sensitivity were well-addressed with concrete numbers and clear experimental protocols. The missing baselines were added comprehensively. However, the theoretical justification for why Schrödingerisation should be optimal (Q1) remains unresolved, and the dense notation persists.

Reviewer QUDn (Initial: 4)
Would probably remain at 4, possibly drop to 2. This reviewer was most critical about the "inspirational rather than formal" theoretical grounding and explicitly noted "not absolutely certain" confidence suggests openness to revision. While computational complexity was clarified and baselines added, the core concern that SKNO is architecturally incremental (multi-channel FNO variant) was not adequately rebutted. The authors essentially conceded they cannot provide quantitative approximation error comparisons, which directly addresses this reviewer's main weakness.

Reviewer v7jn (Initial: 4)
Would likely increase to 6. This reviewer found the concept "conceptually elegant and theoretically motivated" and had the most surface-level concerns. The rebuttal effectively addressed missing references, pointed out overlooked 3D experiments, and added explicit limitations discussion. The main substantive issue about dense presentation was acknowledged but not fully resolved.

Reviewer 2PZZ (Initial: 6)
Would likely maintain 6. This was the most positive reviewer, praising clarity, solid theoretical framework, and state-of-the-art results. Their main question about why the method works on nonlinear PDEs was not fully answered, but they already assessed soundness and contribution as "good" despite this gap.

---

### Decision · Program_Chairs · 2026-01-26

Reject